

# Dissolved Organic Carbon Driven by Rainfall Events from a Semi-arid Catchment during Concentrated Rainfall Season in the Loess Plateau, China

Linhua Wang[1], Haw Yen[2], Xinhui E[1,3], Liding Chen[1,3], Yafeng Wang[1,3,4*]

[1] State Key Laboratory of Urban and Regional Ecology, Research Center for Eco-Environmental Science, Chinese Academy of Sciences, Beijing 100085, China

[2] Blackland Research and Extension Center, Texas A&M AgriLife Research, Texas A&M University, Texas, 76502, USA

[3] College of Resources and Environment, University of Chinese Academy of Sciences, Beijing 100049, China

[4] Institute of Tibetan Plateau Research, Chinese Academy of Sciences, Beijing 100101, China

*Corresponding author: Yafeng Wang: yfwang@rcees.ac.cn*

**Abstract:** Dissolved organic carbon (DOC) transported by runoff has been identified as an important role of the global carbon cycle. Despite there being many studies on DOC concentration and flux, but little information is available in semi-arid catchments of the Loess Plateau Region (LPR). The primary goal of this study was to quantify DOC exported from a sequence of runoff events during the concentrated rainfall season. In addition, factors that affect DOC export from a small headwater

catchment will be investigated accordingly. Runoff discharge and DOC concentration were monitored at the outlet of the Yangjuangou catchment in Yanan, Shaanxi Province, China. The results showed that DOC concentration was highly variable (1.91-34.70 mg L$^{-1}$), with event-based DOC concentrations ranging from 4.08 to 15.66 mg L$^{-1}$. The mean monthly DOC flux loading from the catchment was 94.73-110.17 kg km$^{-2}$ from June to September, while the event-based DOC flux ranged from 0.08 to 2.81 kg km$^{-2}$. Intra-events of rainfall amount and runoff discharge led to event-based/monthly differences in DOC

concentration and flux. Hysteresis analysis showed a nonlinear relationship between DOC concentration and discharge in the runoff process. Our results highlighted the advantages of high-frequency monitoring for DOC export and indicated that DOC export from a catchment is largely influenced by the interaction of rainfall and antecedent conditions for a rainfall event. Engineering and scientists can take advantage of the derived results to better develop advanced field monitoring work. In addition, release of DOC runoff can take quantified during hydrological and biogeochemical processes within catchments in

LPR.

## 1. Introduction

Dissolved organic carbon (DOC), often defined as the solute filtered through <0.45μm pore size, is regarded as one of the active constituents and provides a biologically available carbon source for organisms (Raymond and Saiers, 2010). Thus, the loading of DOC from catchments is widely recognized as a large flux from the soil carbon pool, and is a critical carbon biogeochemical

process in terrestrial and aquatic ecosystems (Battin et al., 2008; Raymond et al., 2013; Raymond and Saiers, 2010). The annual DOC input from terrestrial to ocean environments is approximately 0.25 Pg C, and the release as carbon dioxide from global surface water is estimated at 0.65-3.2 Pg C (Drake et al., 2018; Hedge et al., 1997; Ran et al., 2018). For instance, high DOC concentrations can lead to water pollution and eutrophication, and thus have dramatic consequences on aquatic ecosystem services (Evans et al., 2005; Hu et al., 2016). In addition to ecological impacts, high DOC concentrations will aggravate the

complexation and adsorption of pesticides and heavy metals in hydrological processes. Therefore, the quality of domestic water could be damaged and it might potentially lead to adverse impacts on human health, such as increased risk of cancer, diabetes, or other diseases (Bennett et al., 2009; Ritson et al., 2014). Consequently, the increasing magnitude of DOC via runoff on a global



scale acts as one of the crucial nodes of linking between terrestrial and aquatic ecosystems, and plays an important role in social well-being. Therefore, it is urgent to improve the associated knowledge on DOC concentration variability and develop a
mechanistic understanding of DOC export from catchments.

DOC exported from catchments has attracted great attention in the last two decades due to global concerns about potential influences on the soil carbon pool, aquatic environments and climate change (Laudon et al., 2004; Raymond et al., 2013). For instance, Clark et al. (2007) found that DOC concentration from a peatland catchment was highly variable in storm events (approximately 5-35 mg $L^{-1}$), and a study by Blaen et al. (2017) showed that the DOC concentration ranged from 5.4 to 18.9 mg
$L^{-1}$ with a dynamic DOC source zone. Ma et al. (2018) examined the DOC concentrations in the Three Rivers Headwater Region of the Qinghai-Tibetan Plateau. The results showed that the mean DOC concentration was 3.95 mg $L^{-1}$ and varied with land cover in the catchment. Similar results were reported by Ran et al. (2018), who found that DOC concentration ranged from 1.4 to 9.5 mg $L^{-1}$ in the Wuding River in the Loess Plateau Region (LPR). Previous studies have shown that the release of DOC concentrations from the global catchment ranged from 0.5 to 50 mg $L^{-1}$ and were generally measured at monthly, weekly or daily
intervals (Mulholland, 2003). Such studies highlighted that DOC export was governed by hydrological and carbon biogeochemical processes, precipitation, soil type, and land use (Billett et al., 2006; Dawson et al., 2002; Inamdar et al., 2006). Precipitation can play an important role in DOC export from a catchment. During a rainfall event, increased discharge can caused a higher DOC concentration and flux by flushing accumulated soil organic matter and the relationship between discharge and DOC concentrations is nonlinear. Therefore, a hysteresis effect was found in the runoff process, which was helpful for
characterizing the DOC export process in a rainfall event (Blaen et al., 2017; Lloyd et al., 2016a; Lloyd et al., 2016b; Tunaley et al., 2017). Different intensities of rainfall may alter hydrological connectivity or the flow path and lead to a varied DOC source contributing to runoff. Moreover, the intensity and frequency of rainfall events not only influenced the current hydrological and DOC loading processes, but also changed the soil moisture conditions. In particular, DOC concentration may increase due to accumulated soil organic carbon after a dry period (Jager et al., 2009). In addition, variations in the magnitude and frequency of
rainfall are one of manifestations of climate change. Therefore, characterizing DOC export concentrations, fluxes and patterns from a catchment is critical for understanding DOC processes interacting with hydrological and carbon biogeochemical processes, and is beneficial for predicting DOC flux under future climate change circumstances. Compared to tropical and subtropical areas, comparatively few studies have investigated the dynamics and magnitude of DOC export from semi-arid catchments, particularly in the LPR.

The LPR, which has an area of $6.4 \times 10^5$ $km^2$, is situated in the middle reaches of the Yellow River, China, and approximately 90% of the river loading sediment is derived from this region (Tang, 2004). With regard to this fragile environment, the Chinese government has launched some ecological restoration projects since the beginning of this century, such as the 'Grain-for-Green' and 'Natural Forest Protection Project'. With the implementation of these projects, large areas of steep-sloping (higher than 20°) agricultural land was converted to forest, shrub, or grassland, and engineering measures were also
applied to control erosion (Fu et al., 2017). For instance, check dams can retain sediment and also offer flat and fertile land behind the dam (Wang et al., 2011a). These measures have caused the Loess Plateau to experience a substantial change in land use, vegetation cover, soil properties, and geomorphology. Therefore, the associated hydrological process, ecosystem structure and functions may be alternated accordingly (Chen et al., 2007; Wang et al., 2011b; Wei et al., 2014). Consequently, the hydrological and carbon biogeochemical processes, which operate and interact with each other, were dramatically altered (Liang
et al., 2015a; Liang et al., 2015b). Furthermore, the majority of annual rainfall is concentrated between July and September, accounting 60-70% of total annual precipitation (Shi and Shao, 2000). The dynamics of DOC concentration and flux response to rainfall events may differ from other sites in terms of vegetation and hydrological condition in a ecologically-restored catchment.




Thus, attention should be paid to DOC export driven by rainfall events in a period of concentrated rainfall season, which DOC flux determine status in carbon sink or source for ecological restoration catchments. Until this point, no previous studies have attempted to investigate DOC concentration and the associated flux export over timescales consistent with event-based responses from a semi-arid catchment in the LPR.

Previous studies that monitored DOC loading from catchments have generally used traditional sampling methods at daily, weekly or monthly frequencies to characterize temporal DOC variation in annual or seasonal fluxes. In addition, DOC concentration and flux from catchments are not regular parameters in monitoring networks, such as the Chinese Ecosystem Research Network (CERN). The common approach of sampling frequency is in monthly or weekly schedule at this field gauge station. However, it may result in dissolved organic carbon in runoff samples that is degraded by microbial activities (Kieber et al., 2002; Willey et al., 2000). It is also the reason that only few studies are available in exploring DOC concentration and flux in the research area of this study. Furthermore, no previous study has examined DOC export for a sequence of rainfall events from a catchment in the LPR. Continuously monitoring DOC export and hydrological processes in small catchments using event-based sampling methods has been a helpful technique for adequately capturing the temporal variations and has also broadened a comprehensive insight into DOC export fluxes from catchments. The primary goal of this study is to investigate the variation of DOC concentration for rainfall events to understand the magnitude of DOC flux from an ecologically-restored catchment in the LPR. Three objectives are defined: (i) to examine dynamic changes in DOC concentration and flux during an event-based and monthly period during the concentrated rainfall season; (ii) to quantify the relationship between DOC concentration, flux with discharge derived from detailed monitoring data; and (iii) to assess the rainfall, runoff and antecedent factors influencing DOC export from a semi-arid catchment in the LPR.

## 2. Materials and Methods

### 2.1 Site Description

As shown in Figure 1, this study was conducted in the Yangjuangou catchment (N 36° 42', E 109° 31'), which is an Ecological Restoration and Soil and Water Conservation Monitoring Station situated in Yan'an, Shaanxi Province, China. The catchment is located in the secondary tributary of the Yan River Watershed and covers an area of 2.02 km$^2$, ranging in elevation from 1050 to 1295 m above the mean sea level. The topography is characterized by a typical loess hilly and gully topography with a gully density of 2.74 km km$^{-2}$ (Wang et al., 2011b). The climate of this catchment is situated in a semi-arid continental monsoonal climate with an average annual temperature of 9.6℃. This catchment receives a 535 mm of average annual rainfall, which is unevenly distributed throughout the year and 60-70% of annual rainfall concentrated from June to September. The soil is classified as a typical loess with a fine silt texture and is weakly resistant to detachment by raindrops or runoff. Two check-dams were built in the main gully in the 1960s and it are currently filled with sediment and used for agricultural land. Land use is dominated by forest with a mix of shrub, grassland, and arable land. The major forest species are *Robinia pseudoacacia*, *Salix spp.* and *Populus spp.* The area with *Artemisa argyi*, *Stipa Bungeana trin.*, *Bothriochloa ischaemum*, *Lespedezadavurica schindl.*, and *Artemisia sacrorum* are classified as grassland. The major orchards are *Prunus armeniaca L., Malus pumila Mill.*, and *Juglans rejia L.* The major crops are *Setaria italica, Zea may L. Glycinemax (L) Merr. Panicum miliaceum L.* and *Solanum tuberosum* (Fu et al., 2014). The Yangjuangou catchment was chosen to represent an area with altered land use that has implemented the 'Green-for-Grain' and engineering measures.

### 2.2 Field Monitoring and Sampling



To measure the temporal dynamics of DOC, a monitoring station was deployed at the outlet of the Yangjuangou catchment to sample runoff water and monitor discharge. The station was equipped with an ISCO 6712 (Lincoln, NE, USA) peristaltic pump for collecting water samples during a runoff process induced by rainfall events. High-frequency monitoring was carried out in a rainfall event based hydrological process, thus the ISCO was set to acquire samples every 10 min from the first 12 runoff samples and another 12 were sampled every 30 min. The equipment was programmed to monitor runoff discharge by capturing the runoff flow rate. The auto-sampler collects a runoff sample with a volume of 200 ml. The auto-sampler ceased sampling work after 24 samples were collected. Then, the experimenter poured the runoff water into high-density polyethylene bottles that were prewashed with ultra-pure water. The auto-sampler continued to monitor the hydrological process and sample runoff for the next rainfall event. A meteorological station was installed in the center of the catchment, which was away from high trees. It was used to continuously monitor the rainfall characteristics, air temperature, and soil moisture throughout the study period. Rainfall and air temperature were measured every 30 min. The volumetric soil moisture content at 20 cm depth of forestland was measured every 30 min, which accounts for a large proportion of land use. In addition, the aim of hydrological and meteorological factor monitoring was to characterize the temporal changes and represent the catchment conditions (Blaen et al., 2017). Because these factors drive the hydrological and carbon biogeochemical processes in the catchment, these monitoring works may offer another perspective for understanding the runoff and DOC export process within an individual or continuous temporal variation between rainfall events.

### 2.3 Laboratory Analysis

In the Yangjuangou catchment, researchers resided in the field observatory station and treated the samples immediately after a rainfall event to ensure that the DOC in the sampled water did not microbiologically biodegrade (Kieber et al., 2002; Willey et al., 2000). Therefore, 200 ml of the collected runoff water sample was immediately filtered through a 0.45μm membrane into high-density polyethylene bottles and stored in a cooler (4℃) at the field station. Then, the samples were transported to the State Key Laboratory of Urban and Regional Ecology in Beijing for the following analysis. The DOC concentrations were determined by Vario (Elementar, Germany), which included a high-temperature combustion furnace, a self-contained acidification module and a highly sensitive $CO_2$ detector. Prior to measurement, the instrument should dosed 125 ml of 1% $H_3PO_4$ solution (phosphoric acid) in the acidification module, and validation was then conducted by analyzing various concentrations of a standard solution to achieve accurate results. Ultra-pure water was also tested every 50 samples as a blank to ensure the quality of the results.

### 2.4 Data Analysis

In the present study, the DOC concentration in a runoff event was the flow-weighted mean concentration. The calculation of the flow-weighted mean concentration and flux were defined in the following equations:

$$DOC = \frac{\sum_{i=1}^{n} C_i \times Q_i}{\sum_{i=1}^{n} Q_i} \qquad (1)$$

$$Flux = \frac{DOC \times \sum_{i=1}^{n} Q_i}{S} \qquad (2)$$

where, $Q_i$ (mm) is the discharge corresponding to each sample; $C_i$ (mg L$^{-1}$) is the DOC concentration in an individual runoff sample; n is the number of runoff samples in a runoff event or in a month and $Flux$ (kg km$^{-2}$) is the loading flux of DOC in a rainfall event or the monthly measurement for in the study region; and, $s$ is the catchment area (km$^2$). To better understand DOC concentrations and fluxes from a catchment, specific hydrological and meteorological variables were selected. For instance,





rainfall and soil moisture content may be related to hydrological connectivity in a runoff event, while soil moisture and temperature conditions impact on soil organic carbon content through biological processes (Blaen et al., 2017; Cooper et al., 2007; Soulsby et al., 2003). These variables are Q (total discharge volume a rainfall); RA (total rainfall amount in a rainfall event); R1, R7 and R14 (total rainfall amount in the 1, 7 and 14 days before the current rainfall event, respectively); SMC-7 and SMC-14 (soil moisture content in the 7and 14 days before the current rainfall event); $T_{air}$-7 and $T_{air}$-14 (mean air temperature in the 7and 14 days before the current rainfall event) and REI: interval days between the current and last rainfall event. To analyze potential relationships among DOC concentration, flux, and selected variables, Pearson's test was performed using SPSS (Statistics Package for Social Science). The corresponding figures were developed using Sigma Plot 10.0 (Systat, 2008).

## 3. Results

### 3.1 Rainfall and Discharge in the Study Catchment

Rainfall is the main driving force of hydrological process in a catchment. It is also an important factor affecting the carbon biogeochemical process, which results in substantial temporal changes in the concentration and flux of DOC export from a catchment. Figure 2-a shows the event-based rainfall amount and daily runoff discharge from the June to September, 2016. An event-based rainfall amount measured at the rainfall gauge varied from 56.4 mm (18 July) to 0.60 mm (17 August). The monthly rainfall amounts were 91.2, 192.0, 44.6, and 66.6 mm from June to September in this study catchment (Table 1). Over this period, the total rainfall amount was 394.4 mm, with approximately 74 % of the annual rainfall amount. All the rainfall events in June to September were grouped into four grades: <5 mm (Light rainfall), 5-10 mm (Moderate rainfall), 10-20 mm (Heavy rainfall), and >20 mm (Violent rainfall) according to Yang et al. (2018) classification. Figure 3-a showed that the total rainfall amount was 39.6, 59.4, 104.6, and 187.2 mm for each grade, respectively. The occurrence frequency of rainfall in each grade was 54.9 % (<5 mm), 17.1 % (5-10 mm), 15.7 % (10-20 mm), and 11.8 % (>20 mm) (Figure 3-b). These results indicated that the light and moderate rainfall occurs frequently with a less total rainfall amount, whereas the majority of rainfall amount occurs with a less chance in violent rainfall. In addition, the total amount of sampled events were 19.4, 30, 32.6 and 165 mm for each grade, respectively (Figure 3-a). Consequently, the majority of rainfall occurred from June to September, which is the typical rainfall characteristic of the study area.

For runoff discharge, the mean daily discharge at the outlet of the catchment was 0.46 L s$^{-1}$, but it was also more variable and ranged from 0 to 4.5 L s$^{-1}$. The mean runoff discharges in June-September were 0.35, 0.41, 0.53 and 0.57 L s$^{-1}$, respectively (Table 1). In particular, there was no runoff in the catchment, due to the higher temperature, evapotranspiration and lower rainfall amount in early July. The higher runoff discharge is caused by continuous heavy rainfall. For instance, the cumulative rainfall amount was 91.8 mm and the mean daily discharge flow was 4.05 L s$^{-1}$ from 18-19 July. Similarly, from 15-16 August and 9-10 September, the cumulative rainfall amounts were 45.8 and 28.6 mm. Therefore, the runoff discharge increased rapidly with short duration and violent rainfall. In addition, the evapotranspiration of vegetation decreased as the temperature decreased, and the crops were harvested in the dam field in the late September. The concentrated rainfall season was the source of effective replenishment of the soil water in the dam field. Therefore, runoff discharge still showed a trend of gradually increase slowly with less rainfall occurring during this period. The monthly runoff discharge to rainfall amount ratios were varied from 5.6 % (July) to 31.2 % (August). On average, the discharge to rainfall ratio was 17.2 % in the study period of June to September. In general, runoff discharge tended to follow the pattern of rainfall amount in the study catchment.

### 3.2 DOC Concentrations in Runoff Discharges



### 3.2.1 Event-based DOC Concentrations during Concentrated Rainfall Season

DOC concentrations in runoff discharge were often between 1.91 and 34.70 mg L$^{-1}$. However, the mean DOC concentration for
an individual rainfall event ranged from 4.08 to 15.66 mg L$^{-1}$. Over the study period, patterns in monthly flow-weighted mean DOC concentration were less variable during June to September (Table 1). In general, the monthly mean DOC concentration tended to decrease from 11.52 mg L$^{-1}$ in June to 6.81 mg L$^{-1}$ in August and then slightly increased to 7.49 mg L$^{-1}$ in September. Indeed, the mean DOC concentration was not substantially different during the study period. The relationship between daily discharge and event-based DOC concentration for sampled rainfall events is shown in Figure 4-a. DOC concentrations exhibited
a poor relationship with daily discharge for the Yangjuangou catchment. In addition, the DOC concentration was a more variable and had low runoff discharge compared to the high runoff discharge period, which is typically observed during consecutive rainfall events with high rainfall amount. These results showed that different runoff discharge conditions may affect DOC concentration.

### 3.2.2 Dynamic Changes of DOC Concentrations in a Rainfall Event

To examine the relationship between DOC concentration and runoff discharge in the hydrological process, four rainfall events in each month were selected. Figure 5 shows the dynamic changes in DOC concentration and runoff via the hydrograph over a rainfall event. In general, DOC concentrations varied during the runoff discharge process. The DOC concentration increased quickly in the rising limb of the hydrograph and the maximum concentration occurred behind the peak of the hydrograph on 7 June (Figure 5-a) and 2 August (Figure 5-c), a period with less rainfall and of a long duration. Then, the DOC concentration then
decreased from 1.35 to 0.41 mg L$^{-1}$ at the falling limb on 2 August, while the DOC maintained relatively high values at 1.4-1.50 mg L$^{-1}$ in the falling limb on 7 June. In rainfall events on 13 July (Figure 5-b) and 10 September (Figure 5-d), the discharge hydrograph exhibited a higher fluctuation due to the higher rainfall amount and short rainfall duration. The DOC concentration was kept relatively stable despite the fact that it increased from 1.05 to 1.30 mg L$^{-1}$ at the rising limb on 13 July. However, the DOC concentration sharply increased from 0.61 to 1.24 mg L$^{-1}$ and the maximum DOC concentration was observed before the
peak of the hydrograph. The DOC concentrations then declined and remained stable ranging from 0.61 to 0.75 mg L$^{-1}$ at the falling limb on 10 September. Overall, the dynamic changes in DOC concentrations in the hydrograph show that the DOC export process varied with different rainfall and runoff conditions.

A hysteresis analysis was used to examine the dynamic changes of the DOC concentration response to a hydrological process, which has been applied to investigate the temporal variation in concentration export for a catchment. Figure 6 shows
that the DOC concentrations varied in the rising and falling hydrograph during four selected rainfall events (7 June, 13 July, 2 August and 10 September 2016). Three hysteresis patterns were observed, including clockwise (13 July and 10 September), anti-clock wise (7 June) and figure of eight (2 August). As shown in Figure 6-a, the DOC concentrations were higher during the falling limb than during the rising limb of the hydrograph, thus resulting in an anti-clock wise pattern. The delayed maximum DOC concentration may be attributed to longer runoff flushing upstream, with increased hydrological connectivity induced by
longer rainfall duration. Thus, it leads to a relatively higher DOC concentration during the falling period. Figure 6-c showed a figure-of-eight pattern and indicated that DOC concentration generally varied in pace with runoff discharge on 2 August, 2016. The difference of DOC concentration between rising and falling limb at a given flow rate was small, as supported by the results shown in Figure 5-c. On 13 July (Figure 6-b) and 10 September (Figure 6-d), the DOC exhibited a clockwise pattern, which implied that the DOC concentration was higher in the rising limb than in the falling limb. The increased DOC was attributed to
rapid flushing from soil organic carbon into stream water during the rising limb, and the DOC then declined due to the dilution effect. Moreover, previous studies reported that the close link between the DOC source and discharge flow may lead to a rapid




increase in the DOC concentration. The relationships between concentrations in monthly and event-based processes highlighted that the DOC export behavior was different in a complete hydrological process or intra-events.

### 3.3 DOC Fluxes from Catchment

A rainfall event-based monitoring method is helpful to better understand the hydrological, DOC concentration and flux process. The rainfall event based on DOC flux ranged from 0.08 to 2.81 kg km$^{-2}$ with a mean DOC flux of 0.43 kg km$^{-2}$ for all sampled rainfall events from June to September 2016. The relationship between event-based DOC flux and runoff discharge amount is shown in Figure 3-b. The DOC flux showed a positive linear relationship with the runoff discharge amount for all sampled rainfall events. However, the DOC flux was more variable in lower runoff discharge conditions. For the monthly DOC flux, the

total DOC loading from the catchment ranged from 94.73 kg km$^{-2}$ in August to 110.17 kg km$^{-2}$ in September (Table 1). Although the total runoff discharge was lowest in June in these four months, the DOC monthly flux was 102.39 kg km$^{-2}$ and had a higher flow-weighted DOC concentration (11.52 mg L$^{-1}$). However, the DOC flux was higher in September, with an increased runoff discharge and a lower flow-weighted DOC concentration. The larger runoff discharge amount may offset the effects of lower DOC concentrations. These results showed that variation in DOC flux during sequential rainfall events induced hydrological

processes in the concentrated rainfall season. Thus, it highlights the complexity of the DOC loading process, which may be influenced by rainfall, hydrological and carbon biogeochemical processes.

### 4. Discussion

### 4.1 Relationship between Rainfall and DOC Export

It has been known that hydrological and carbon processes are important aspects of the regional carbon cycle and for restoring
ecosystem service. However, the DOC exported from a catchment in the LRP has rarely been studied. In this study, we uses an in-situ auto- and high-frequency monitoring method to observe temporal changes in hydrological and DOC concentration dynamics for an event-based sampling period during the concentrated rainfall season (June-September) (Figure 2-b). For DOC export on a monthly scale, the DOC was calculated as the product of total discharge and flow-weighted mean concentration in a month; and thus, these two variables represented hydrological and carbon biogeochemical processes. Monthly DOC fluxes were

not clearly correlated with discharges. The flow-weighted DOC concentrations decreased during the experimental period, which differed from the greater DOC flux with a large discharge (Chen et al., 2012; Cooper et al., 2007). Furthermore, the monthly DOC fluxes were negatively correlated with the discharge amount from June to August 2016. The DOC concentration was higher in June and decreased in August. This was reasonable because the accumulated soil organic carbon can be flushed by runoff in early rainfall period, and the DOC concentration may be diluted by increased runoff (Blaen et al., 2017; Chen et al.,

2012). In addition, in combination with the increased discharge amount, the decreased concentration led to a decrease in monthly DOC flux from June to August. This could be explained by the relative changes in DOC concentrations being higher than changes in monthly discharge, indicating that the decreased concentration may outweigh the effect of increased discharge. However, the exception occurred in September, while increased DOC flux over the other three months was mainly due to a smaller increase in DOC concentration. These results were also probably associated with rainfall amount, land cover and runoff

flow path (Laudon et al., 2004; Soulsby et al., 2003). For example, crops planted in the check-dam field were harvested, and the ratio of rainfall to runoff increased in September. The soil soluble organic carbon is more likely to leach through macropores from check-dam farmland into runoff, which further increased the DOC concentration in runoff. Thus, it led to a slight increase





in DOC flux in September. Therefore, it could be inferred from these results that DOC flux may depend on runoff flushing capacity and flow path in a restored and check-dam catchment.

Despite the fact that the DOC fluxes varied in different months, there were also differences in DOC concentration and flux response to a rainfall event. Figure 7 showed the relationship between rainfall amount and mean daily discharge during June to September. This indicated that a rainfall event-driven discharge varied with rainfall amount, and thus need to be grouped by rainfall amount according to the rainfall grade. Linear regression analysis between discharge and rainfall amount with larger than 20 mm ($k$=0.07, $R^2$=0.62) showed a more rapidly changes than with rainfall amount less than 20 mm ($k$=0.02, $R^2$=0.59). Our

results suggested that runoff discharges are highly sensitive to larger rainfall amount with greater than 20 mm in this area. In combined with soil moisture content changes during monitoring period, the infrequent and amount of violent rainfall events strongly influence the discharges, DOC during or later export from a catchment. Therefore, rainfall events caused considerable variations in event-based DOC concentration (Figure 4). There is a poor relationship between discharges and DOC concentrations for all sampled rainfall events and means that a necessary to grouped by rainfall amount correspondingly. Weak

and negative relationship were observed between DOC concentrations and runoff discharges induced by all rainfall events less than 20 mm ($R^2$=0.002, N=37). In contrast, DOC concentration generally decreased with discharge and showed a clear negative correlation for rainfall events higher than 20 mm ($R^2$=0.038, N=5). These findings are contrary to the studies that larger discharges resulted in higher DOC concentrations (Hope et al., 1994; Ma et al., 2018; Williams et al., 2017). Indeed, the large variations in DOC concentration were observed a general dilution effect induced by higher rainfall amount with larger runoff

discharge, as reported by Clark et al. (2007) and Evans et al. (2006). Our results suggested that a lower discharge induced by lower rainfall amount have a more complex and larger influence on DOC concentration in semi-arid catchment of Loess Plateau.

An event-based DOC flux is a function of total runoff discharge and DOC concentration. DOC flux showed a positive linear relationship with runoff discharges, which is not surprising and parallel with studies reported by Clark et al. (2007) and Ma et al. (2018). Although the DOC concentration decreased with the runoff discharge, the greater DOC flux clearly showed that the

DOC export from a catchment was linked to hydrologic process induced by various amount of rainfall events. Therefore, these results showed that rainfall characteristics, hydrological conditions and soil carbon content are important for understanding DOC export from a catchment.

### 4.2 Factors Influence on DOC Export from a Catchment in Semi-arid Region

The mechanisms of DOC export from terrestrial ecosystems may be complicated and depend on many factors, such as soil

organic carbon, vegetation, rainfall, hydrological condition and sampling period (Blaen et al., 2017; Cooper et al., 2007; Ma et al., 2018). Comparatively few previous studies have investigated how changes in hydrological factors and rainfall affect on DOC export. For instance, a current rainfall event leads to changes in a hydrological process, and it may also simultaneously change soil moisture content, which may influence the soil carbon biogeochemical process. For the next rainfall event, the antecedent conditions, such as hydrological conditions and the soil organic carbon content, may also influence the DOC concentration and

flux. In general, antecedent conditions drive DOC export through exerting influences on availability of DOC and impacts on hydrologic connectivity (Brocca et al., 2010; McMillan et al., 2018). Therefore, DOC export from a catchment during rainfall events was the result of carbon biogeochemical processes, and the antecedent hydrological and rainfall characteristics. In this study, temporal variations of rainfall, air temperature and soil moisture content were continuously monitored throughout the study period to provide detailed information describing the antecedent and current conditions.

Table 2 showed the correlation between DOC concentration/flux and a set of factors in all sampled rainfall events during the study period. The event-based DOC concentrations were positively correlated with rainfall amount (RA) and R7. These



results suggest that the combination of the current rainfall amount and the accumulated rainfall before a current rainfall event are important. The influence of R7 may reflect the antecedent hydrological condition that impacted runoff generation and connectivity in a rainfall event rather than the direct influence on DOC concentration. However, event-based DOC concentrations were extreme significantly and negatively correlated with SMC7 and SMC14. A previous study by Yang et al. (2018) in the LPR found that rainfall recharge into soil had a rainfall threshold value. The soil moisture content was continuously dried and then effectively rewetted under a specific rainfall amount, as supported by the soil moisture variations shown in Figure 2-c. Given variation pattern of soil moisture content during studied period, the relationship between DOC concentration and the dry-wet cycle of soil moisture was shown in Figure 8. The higher DOC concentrations from June to middle July coincided with light rainfall and low discharge. This is probably attributed to inactive microbial activity, caused by the relatively lower soil moisture (Jager et al., 2009). The DOC concentration decreased with increased soil moisture content, particularly in July 18 with a total rainfall amount of 56.4 mm. On one hand, violent rainfall events may induce a higher discharge, causing a dilution effects on DOC concentration. On the other hand, the rainfall water may effectively replenish soil moisture content, and thus stimulate a higher decomposition of soil carbon under wet and higher temperature condition. Then, the relative decreased DOC concentrations were observed in a drying soil moisture condition for the next rainfall events, which may attribute to an exhaustion of DOC (Laudon et al., 2004). These findings were similar to previous studies by Tunaley et al. (2017), who reported a strong influence of dry antecedent conditions on DOC export response to rainfall event. For event-based DOC fluxes, DOC flux was significantly and positively correlated with Q, RA, R1 and R7. The Q and RA reflect the direct effect of current rainfall and hydrological processes during a rainfall event, while R1 and R7 refer to the antecedent rainfall conditions and reflect indirect effects on DOC export. These results agreed with previous studies demonstrated by Blaen et al. (2017), who noted that antecedent conditions and rainfall were key drivers of DOC export during a rainfall event. Cooper et al. (2007) also concluded that DOC export is largely governed by interactions between hydrological and meteorological factors and carbon biogeochemical process. Since the DOC export process is complicated, the influence of the selected factors, including current and antecedent conditions, is intuitive to some extent. Therefore, prediction of DOC concentration induced by rainfall is complicated to establish. Given that our results were based on rainfall event date and collected during a concentrated rainfall season in a small catchment, this process requires further investigation regarding soil organic carbon changes under intra rainfall events and dry-rewetting conditions on DOC release from soil into runoff during different rainfall events in the LPR.

## 5. Conclusion

The DOC concentration and flux for individual rainfall events from a semi-arid catchment of the LPR was initially monitored during the concentrated rainfall season. DOC concentration showed a weak correlation with discharge, except in higher runoff discharge induced by extreme rainfall events. DOC flux increased with runoff discharge and showed a positive linear correlation with runoff discharges. These results showed that higher DOC export with low DOC concentration related to higher discharge and its dilution effects, in turn caused by larger rainfall amount. The diluted DOC concentration induced by increased discharges contributed slightly to difference in DOC flux, due to total runoff discharge is a major variable for flux. The findings of this study indicate that DOC concentrations were highly variable, particularly during low runoff discharge periods. The detailed monitoring method used to capture multiple factors, including runoff discharge, rainfall, DOC concentration, soil moisture and temperature through the concentrated rainfall season, facilitates a better understanding of the dynamic DOC export process in a rainfall event. Hysteresis analysis showed that the relationship between DOC concentration and runoff discharge for a rainfall event is nonlinear and varied with conditions in rainfall amount, discharge process, and monitoring time. These results showed that the DOC response to rainfall is related to hydrological conditions (Q and RA) and antecedent conditions (R1, R7 and SMC).



Therefore, our results preliminarily highlight that DOC export was influenced by the interaction of hydrological and carbon biogeochemical processes. Engineers and scientists can take advantage of the derived results to better develop advanced field monitoring work. In addition, release of DOC in runoff can the quantified during hydrological and biogeochemical processes within catchments in LPR.

*Data availability*: The dataset used for this manuscript can be provided by e-mail contact with the first or corresponding author.

*Author contributions*. Linhua Wang: analyzing data and organizing the manuscript; Haw Yen: discussing the relationship between DOC concentration/flux and runoff discharges induced by a sequence of rainfall events; Xinhui E: sampling and lab testing work; Liding Chen ang Yafeng Wang: discussing and guiding the field monitoring work.

*Competing interests*: The authors declare that they have no conflict of interest.

**Acknowledgments**

This work was financially supported by the National Nature Science Foundation of China (41671271, 41807176 and 41571130083). We wish to thank Shibo Chen and Weiliang Chen for their help during the field sampling work and laboratory analysis.

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

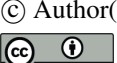



Table 1 Characteristics of rainfall, runoff discharge, DOC concentration and flux during the concentrated rainfall season (June-September, 2016).

Table 2 Summary of correlation coefficients between DOC concentration/flux and a set of factors.

Figure 1 Geographic location of the Yangjuangou catchment in the Loess Plateau Region, China and the red and yellow dot denote the weather station and runoff sampling site.

Figure 2 Temporal variations in rainfall amount (a), runoff discharge (a), DOC concentration and flux (b), soil moisture content (c) and air temperature (d) during the concentrated rainfall season (June-September, 2016). The green dots and bars denote the sampled rainfall events.

Figure 3 Statistics of rainfall events from June to September, 2016: (a) statistics of total and sampled rainfall amount, (b) statistics of rainfall grades and its occurrence frequency.

Figure 4 Relationships between runoff discharge and DOC concentration and flux for sampled rainfall events during the concentrated rainfall season (June-September, 2016).

Figure 5 Dynamic changes of DOC concentration in an individual runoff event: (a) 6 June, 2016; (b)13 July, 2016; (c) 2 August, 2016; (d) 10 September, 2016.

Figure 6 Hysteresis loops for four selected runoff events from June to September: (a) 6 June, 2016; (b)13 July, 2016; (c) 2
August, 2016; (d) 10 September, 2016.

Figure 7 The relationship between rainfall amount and discharges during monitoring period.

Figure 8 The relationship between DOC concentration and dry-wet variation of soil moisture concentrated during monitoring period.







**Table 1**

| Month | Rainfall amount (mm) | Mean discharge (L s$^{-1}$) | DOC concentration (mg L$^{-1}$) | | Monthly flux (kg km$^{-2}$) |
|---|---|---|---|---|---|
| | | | Range | Mean | |
| June | 91.2 | 0.35 | 9.97-15.66 | 11.52 | 102.39 |
| July | 192.0 | 0.41 | 6.70-13.00 | 8.95 | 96.57 |
| August | 44.6 | 0.53 | 4.38-9.72 | 6.81 | 94.73 |
| September | 66.6 | 0.57 | 4.08-14.15 | 7.49 | 110.17 |

**Table 2**

| | Flux | Q | RA | R1 | R7 | R14 | REI | $T_{air}$-7 | $T_{air}$-14 | SMC-7 | SMC-14 |
|---|---|---|---|---|---|---|---|---|---|---|---|
| DOC | 0.30 | -0.01 | 0.30 | -0.01 | 0.23 | -0.05 | -0.32[*] | -0.25 | -0.24 | -0.44[**] | -0.65[**] |
| Flux | | 0.94[**] | 0.69[**] | 0.76[**] | 0.57[**] | 0.29 | -0.14 | -0.07 | -0.04 | 0.06 | -0.24 |
| Q | | | 0.60[**] | 0.85[**] | 0.53[**] | 0.33[*] | -0.07 | -0.02 | 0.01 | 0.19 | -0.03 |
| RA | | | | 0.38[*] | 0.39[*] | 0.14 | -0.06 | 0.02 | 0.07 | -0.05 | -0.30 |
| R1 | | | | | 0.58[**] | 0.42[**] | -0.27 | 0.11 | 0.10 | 0.12 | -0.01 |
| R7 | | | | | | 0.69[**] | -0.28 | 0.24 | 0.23 | 0.40[**] | 0.02 |
| R14 | | | | | | | -0.20 | 0.19 | 0.13 | 0.56[**] | .420[**] |
| REI | | | | | | | | -0.02 | 0.03 | 0.26 | 0.25 |
| $T_{air}$-7 | | | | | | | | | 0.96[**] | 0.09 | 0.20 |
| $T_{air}$-14 | | | | | | | | | | 0.09 | 0.17 |
| SMC-7 | | | | | | | | | | | 0.79[**] |

Note: ** (P<0.01), * (P<0.05).




**Figure 1**

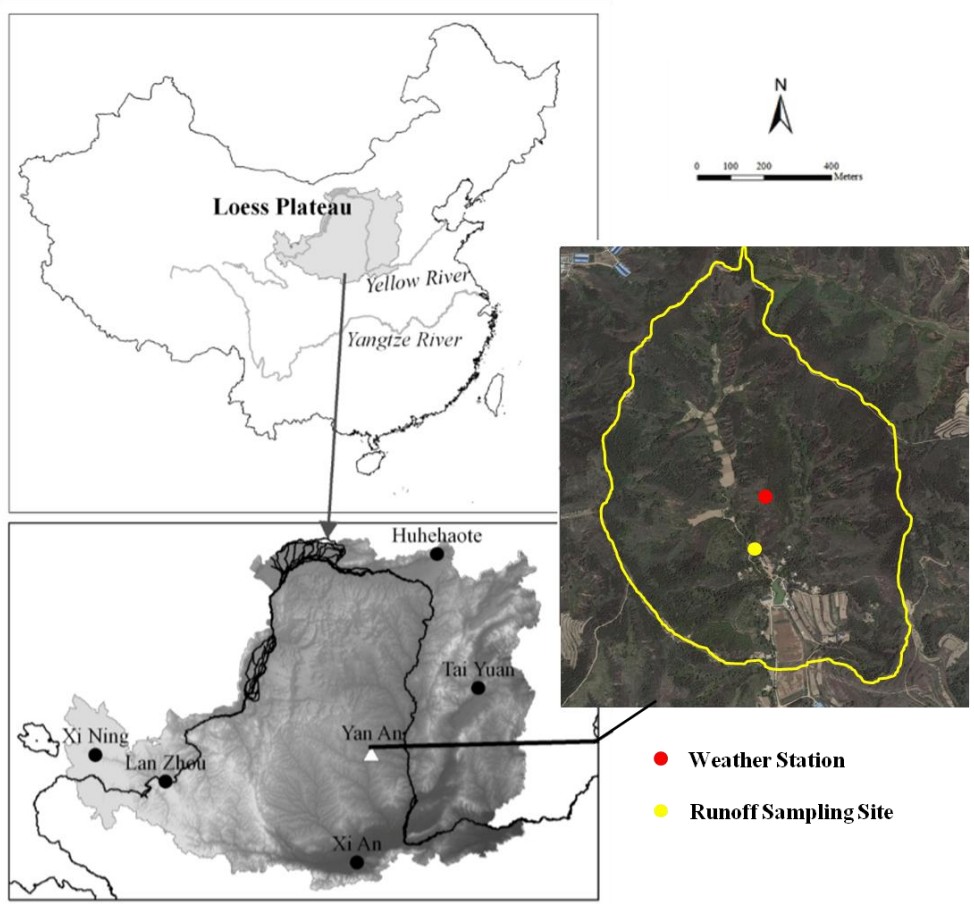







**Figure 2**




**Figure 3**

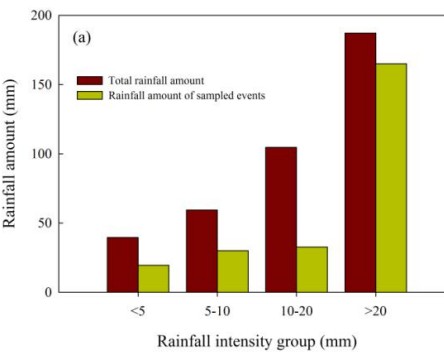
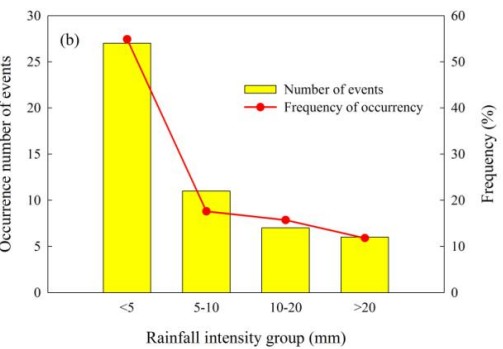

**Figure 4**

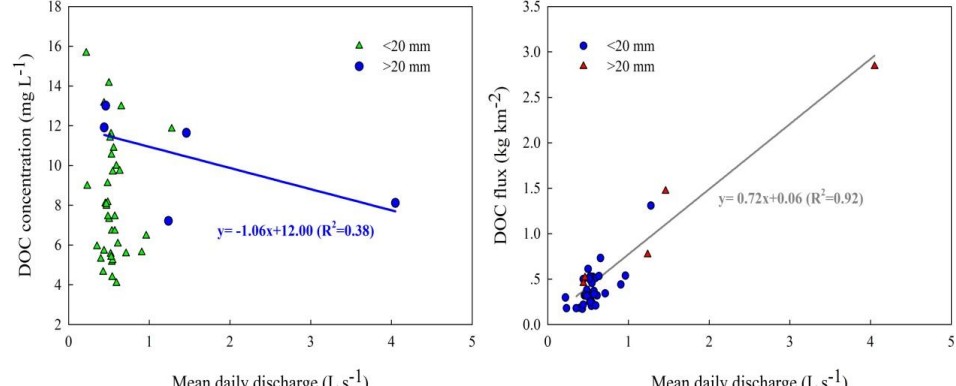






**Figure 5**

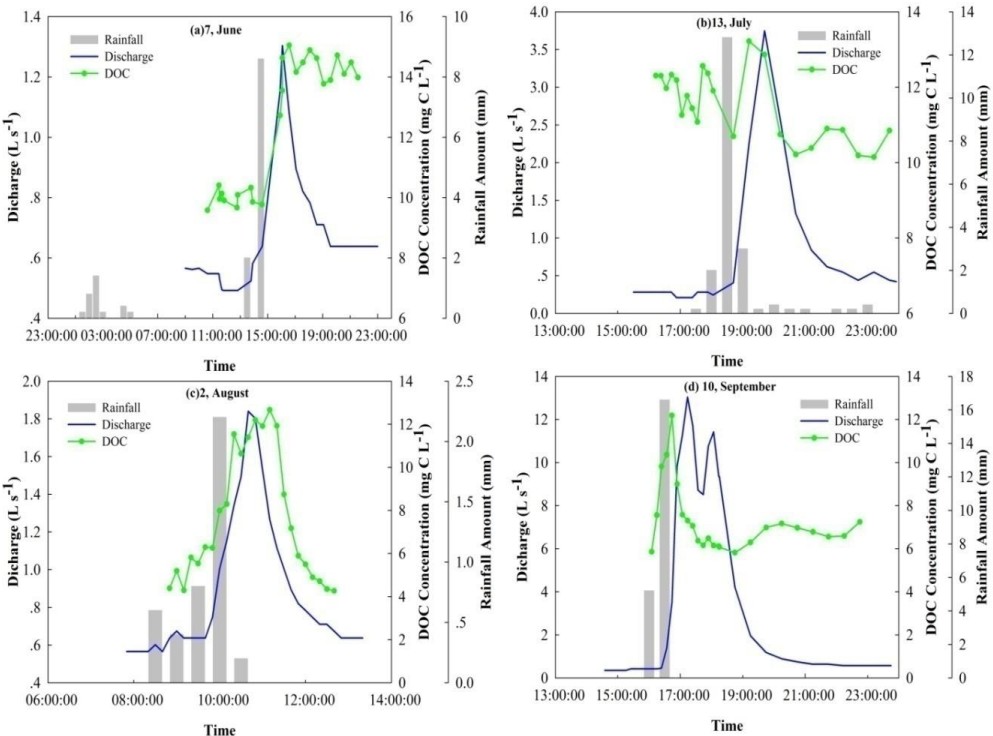





**Figure 6**

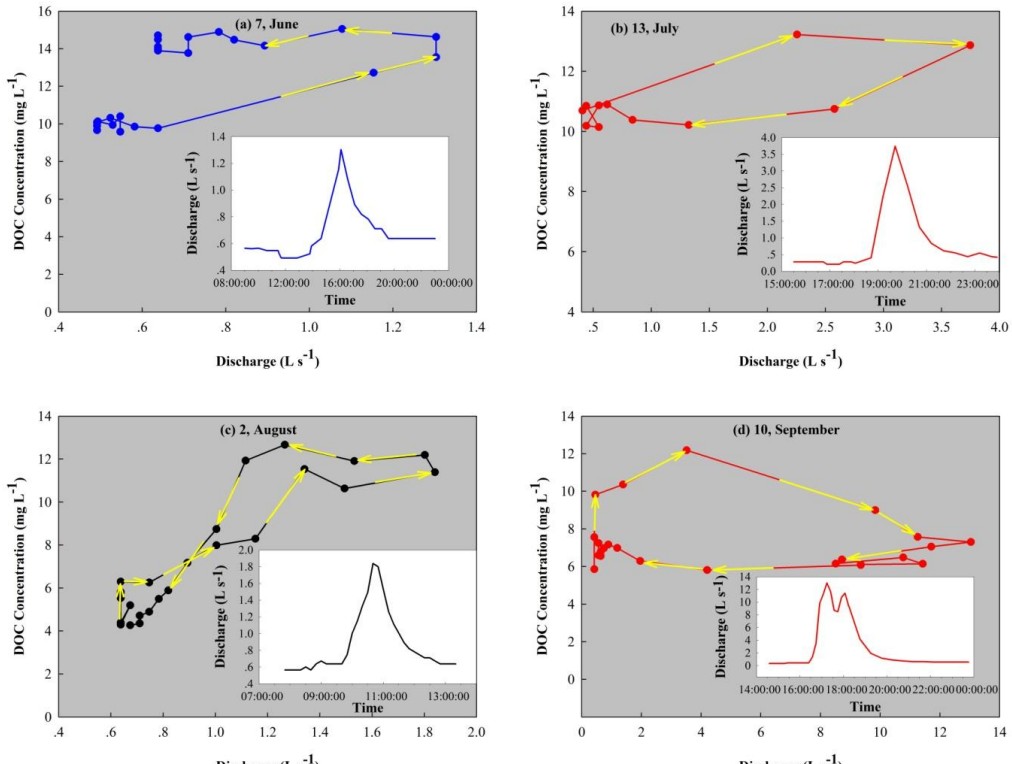







**Figure 7**

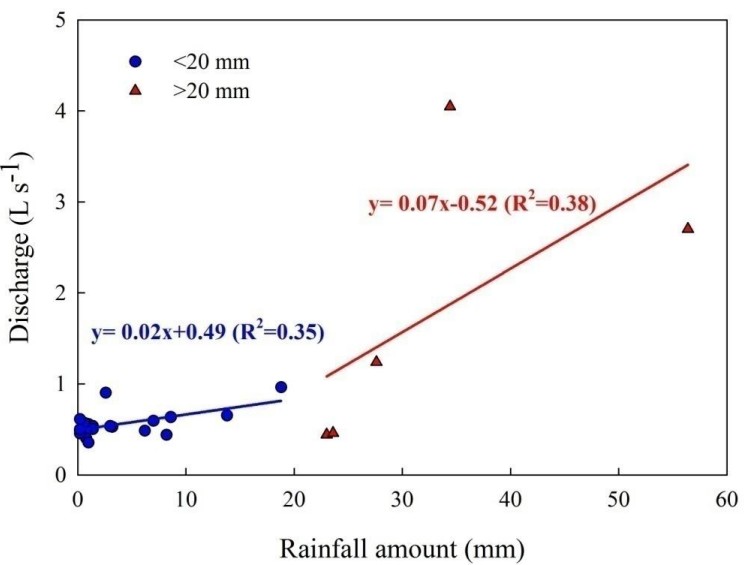


**Figure 8**

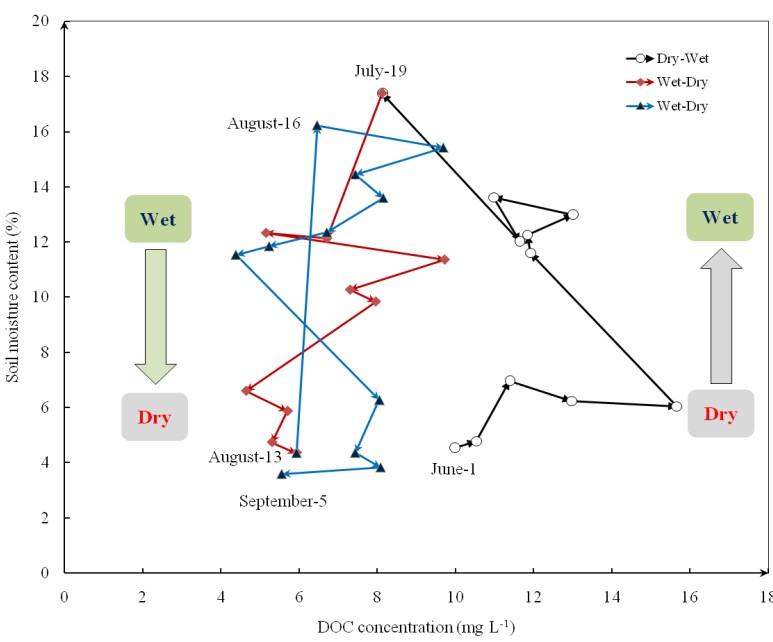