# Peer review of "Dissolved Organic Carbon Driven by Rainfall Events from a Semi-arid Catchment during Concentrated Rainfall Season in the Loess Plateau, China"

_Hydrology and Earth System Sciences, 2019_

## Referee Comment (RC1) · Anonymous Referee #1 · 11 Mar 2019

Dear Editor,

Thank you so much for the opportunity given to evaluate this manuscript. This manuscript discussed dissolved organic carbon transported by rainfall events based on high-frequency monitoring method on the Loess Plateau of China. The study area, Yangjuangou catchment, is a typical watershed of ecological restoration and soil and water conservation which has important scientific meaning for the Loess Plateau. This could contribute to a better understanding of carbon transport in hydrological process at catchment scale. Therefore, I suggest that the manuscript could be considered after a major revision. The detail comments are listed below:

[Figure]

Q1 Page 2, line 41-45;

Please illustrate the detailed potential influences of DOC on the soil carbon pool, aquatic environments and climate change before enumerating the concentration ranges of DOC in different regions.

Q2 Page 2, line 62-64; page 2, line 79-81; page 2, line 88-89;

I think that the lack of the previous paper about DOC export in the Loess Plateau should be presented in a more concise way and it is better to combine the lack of previous research and put forward your own hypothesis.

Q3 Page 3, line 114;

Please introduce the time/period of sampling or monitoring, how much rainfall events were monitored and how many samples were collected in the section of Field Monitoring and Sampling. The specific rainfall events mentioned in 3.2.2 and the reason for selecting these events should also be explained in 2.2.

Q4 Page 4, line 136-140;

Please complete the name of the TOC analyser, like Vario TOC select or Vario TOC cube. I think it would be much better to describe what the 1% $H_3PO_4$ solution is used for.

Q5 Page 4, line 143 and line 147-148; page 18, line 515;

The meaning of DOC concentration and discharge should be consistent throughout the paper. DOC concentration has been defined as the flow-weighted mean DOC concentration. $C_i$ and $Q_i$ were defined as the discharge and DOC in an individual runoff sample. However, the Y-axis in figure 5 corresponds to the discharge and DOC concentration of each sample. It is better to present flow-weighted mean DOC concentration in another way to distinguish it from the DOC in runoff samples and DOC in other studies.

Q6 Page 5, line 153-156;

It would be better to use commas instead of semicolons. There should be a space between "7" and "and" (line 155-156). Do not use a colon to explain abbreviations (line 156).

Q7 Page 5, line 157-158;

Please use the version information of SPSS instead of Statistics Package for Social Science.

Q8 Page 6, line 193-194;

Why do you use the daily discharge instead of rainfall event discharge or flow rate analyse the relationship between discharge/flow and DOC concentration? I think the relationship between runoff and DOC concentration could be better explained by the DOC concentration of each runoff sample and its flow rate in the corresponding sampling period.

Q9 Page 6, line 224-227;

This part is the interpretation or analysis of the above results, and it should be included in the section of discussion with the citation.

Q10 Page 7, line 243; page 8, line 288;

What is the difference between DOC export and DOC flux? DOC export and flux were mixed in several parts of the paper affecting readers' understanding of the study. Please explain the specific meaning of DOC export.

Q11 Page 7, line 245-247;

Substitute "uses" with "used". As shown in figure 2-b, the DOC concentration looked like the average DOC concentration daily or per rainfall event, rather than the results of each collected runoff samples. This figure did not show the high-frequency monitoring in your study.

Q12 Page 7, line 250-251;

Is the result of the greater DOC flux with a large discharge obtained from your study? If not, it is better not to compare the result about DOC flux in previous studies with the result of DOC concentration in your study.

Q13 Page 8, line 266-267, line 268-26, line 274-277 and line 300-301; page 9, line 308-310;

The above sentences are the description of the figures or tables and it is best to move those to the result section.

Q14 Page 8, line 268-269 and line 274-277;

Why do you take the 20mm rainfall amount as the break point to do the linear regression analysis respectively?

Q15 Page 8, line 288;

The subtitle is too broad and general. Rainfall, one of the most important factors affecting DOC concentration, has been mentioned in 4.1, but not been fully discussed.

Q16 Page 16, line 500, Figure 2;

Try to use shading or background fill to distinguish the values of sampling days instead of using different coloured dots.

Q17 Page 16, line 505, Figure 4;

Is the regression curve in the right side of figure 4 fitted according to all sampled rainfall events or according to >20 mm rainfall events? Please indicate (a) and (b) in the figure 4.

---

## Referee Comment (RC2) · Anonymous Referee #2 · 18 Mar 2019

GENERAL COMMENTS:

This manuscript investigates the dynamic changes in DOC concentration and flux and their relationships to rainfall, runoff characteristics and climatic and soil factors in a semi-arid catchment in Loess Plateau Region, China. To reach the goal, they coupled analyses of runoff water at rainfall event-based frequency, with meteorological data continuously monitoring the rainfall characteristics, air temperature, and soil moisture. Their results showed higher DOC export with low DOC concentration, highlighted that DOC export was influenced by the interaction of hydrological and carbon biogeochemical processes.

In general, this manuscript is solid work and worthwhile to publish although the presented results are not very spectacular. The high frequency sampling of runoff (24 samples per rainfall) well revealed the temporal change in DOC concentration within a typical event as they showed in four selected events. However, the author should consider re-evaluate the data and re-organize the writing since the event-based analysis is not well discussed. In my opinion, it will be a good contribution biogeochemistry modelers if author delve into individual rainfall event and give information that clearly linking DOC export to rainfall and runoff characteristics.

MAJOR COMMENTS:

1. Introduction

(1) L43-50: these piled data didn't give a clear background on DOC export. They should be re-organized and present in term of different catchment characteristics. For example, does DOC export differ in area receiving different annual precipitation, or area with different land use, or catchment size and topography.

(2) The knowledge gap is not well stated. I believe the event-based analysis has been done in other region and what kind of information it can provide compare to traditional daily/monthly monitoring? Why do you think you need to conduct event-based analysis in this specific semi-arid area?

2. Materials and Methods

Be more specific about the experiment duration. How long/how many rainfall events have you been monitoring and sampling?

3. Results and Discussion

(1) The author should either consider combining the result and discussion sections OR separating them clearly in the writing. There are multiple places that the results been re-stated in discussion or discussed the result right after without citation.

[Figure]

e.g. L181-L183 should be in discussion section. L196-L197 discussed the result without citation and should be in discussion section. L223-L228 should be in discussion section. L267-269 should be in results section. L274-276 should be in results section. Section 4.2 (L300-L327): results about this part (Table 2) wasn't mentioned in results section.

(2) I'm confused with the way you separate the rainfall events into 4 groups. In Figure 3 you stated in x-axis was rainfall intensity, but the unit was mm, not mm/h. Why do you define rainfall intensity based on accumulated rainfall depth? In the Yang et al. (2018) paper you referred, they denoted the rainfall replenishment in mm to effectively recharge the soil water. I don't think this is the classification you should use here. My recommendation would be to directly calculate the rainfall density for each individual rainfall event.

(3) The major finding you stated was higher DOC export with low DOC concentration. I have several questions about this finding: In Figure 4, you stated DOC concentration depressed with increased discharge for greater intensity. How are you sure since you only have 5 points with r2 value of 0.38. Is this correlation significant? In Figure 5, DOC do show positive relationship with discharge within individual event, how do you explain this contrary?

SPECIFIC COMMENTS:

L32: insert a summary sentence before "For instance, high DOC...". The following statements come from nowhere and it's confusing.

L104: give the time period for average annual temperature and precipitation. Is 535 mm only coming from rainfall or also including other type of precipitation?

L112: state specific land alteration in "represent an area with altered land use that has..."

L126-L130: the part "In addition, the aim of hydrological..." should be stated before

you introduce the meteorological station, and also should be condensed.

L133: "microbiologically biodegrade" to "microbially degrade".

L138-L140: give CV of procedure accuracy.

L149-L156: this section should be in laboratory analysis or an independent section rather in data analysis.

L166: "in June to September" to "in between June and September".

L186: I suggest to open this paragraph with sentence "In general, runoff discharge tended to follow the pattern of rainfall amount in the study catchment."

L189: where did the value "34.70 mg L-1" come from? I didn't see this value in Figure 2 or Figure 5.

L191: "DOC concentration were less variable during June to September", less compare to what?

FIGURES:

Figure 3: see previous comments about the grouping.

Figure 4: Is the second figure necessary? DOC flux is calculated based on discharge. Why present a variable that is highly dependent on the other variable?

Figure 5: explain in result section why did you choose these four events? Do they show different rainfall intensity? Axis of DOC concentration could be in the same scale.

Figure 6: Axis of DOC concentration could be in the same scale.

Figure 7: my recommendation is to compare discharge with rainfall intensity.

---

## Referee Comment (RC3) · Anonymous Referee #3 · 20 Mar 2019

This manuscript reported the changes in DOC concentration and flux and their relationship to rainfall events. Although the authors did some solid fieldwork, I think the authors need to restructure the manuscript to present their scientific findings. I have some major concerns for the authors: 1) The authors declared that there were no reports to the LPR, but I do not think that is the reason they conducted such a study. 2) They claimed that this study highlighted the interaction of rainfall and antecedent conditions for DOC exports in a catchment, but they did not say what interactions and what effects. 2) The introduction is very difficult to follow, they presented a numerous report (for example, L40-L64), I think they need to summary these studies and then the potential readers can know why they design this study. 3) The three objectives of

this study were not well described in the introduction section. Actually, I think the LPR (L65-L81) and CERN is nothing to do with the scientific questions only if the authors can upscale their results to the whole LPR and explained the global significance of the DOC exports in LPR. I believe most readers have no idea about the sampling sites in this study, and they also do not care about this. 4)The result section is too long, making it difficult to read. The authors need to redo the tables and figures. I do not understand Figure 2a. It is also difficult to understand Fig. 8. The authors also need to explain the abbreviations for R1, R2 in table 2 so the readers need not to find them in the text. 5) The authors should clarify the rainfall amount and rainfall intensity, which is important to class the rainfall events. 6)L275-278, it is unclear about the time interval between these sampling times. If they want to conduct such analysis, they should check the original data to ensure the normal distribution. 7)Conclusions. The findings of this study indicate that DOC concentrations were highly variable, particularly during low runoff discharge periods, granted, this belongs to the conclusion. But many other sentences just simply repeated the results. The authors should think hard about the findings of this study and show that these findings are valuable.

---

## Author Comment (AC1) · 11 Apr 2019

**Response to HESS-2019-8-RC1**

**Anonymous Referee #1**

Thanks for your suggestions. We appreciate for anonymous referee comments concerning our manuscript entitled "Dissolved Organic Carbon Driven by Rainfall Events from a Semi-arid Catchment during Concentrated Rainfall Season in the Loess Plateau, China". We have studied comments carefully and have made corrections. The main corrections in the manuscript according to the referee's comments are as follows:

**Comment 1.** Please illustrate the influences of DOC on the aquatic environments and climate change before enumerating the concentration ranges of DOC in different regions.

**Response:** Thanks for your suggestions. Line 33-36 describe the potential influence of DOC.

**Line 33-36:** For instance, high DOC concentrations can lead to water pollution and eutrophication, and thus have dramatic consequences on aquatic ecosystem services (Evans et al., 2005; Hu et al., 2016). In addition to ecological impacts, DOC in runoff also play an important role in social well-beings. High DOC concentrations will aggravate the complexation and adsorption of pesticides and heavy metals in hydrological process.

**Comment 2.** The previous paper about DOC export in the Loess Plateau should be presented in a more concise way and it is better to combine the lack of previous research and put forward your own hypothesis.

**Response:** Thanks for your suggestions. The lack of previous research and objective part has been rewritten and the details show in Line 69-79 of this manuscript:

**Line 69-79:** Less information is available on DOC export driven by rainfall event, which DOC flux is an important component in overall carbon balance for ecological restored catchment.

Therefore, the primary goal of this study is to investigate how variations of DOC concentration and flux response to a sequence of rainfall events from a restored catchment during concentrated rainfall season in the LPR. Specifically, the two objectives of this study were (1) to examine the dynamic changes in DOC concentration and flux and assess the difference in DOC export driven by various rainfall events, and (2) evaluate how rainfall, runoff, and antecedent factors affect DOC export from a catchment. To do so, we used high-frequency method to capture the temporal changes in DOC export and hydrological process driven by rainfall event within an ecological restored watershed in LPR. These results will provide evidence of DOC export response to rainfall events, especially driven by extreme events, which may be important for evaluating carbon balance and modeling DOC export through runoff at ecological restored catchment in LPR.

**Comment 3.** Please introduce the time/period of sampling or monitoring, how much rainfall events were monitored and how many samples were collected in the section of Field Monitoring and Sampling. The specific rainfall events mentioned in 3.2.2 and the reason for selecting these events should also be explained in 2.2.

**Response:** Thanks for your suggestions. Line 108-109 added the sampling information and Line 184-186 added the reason for selecting rainfall events.

**Line 108-109:** There were 278 samples collected for 22 hydrological processes induced by rainfall event over the monitoring period of June to September, 2016.

**Line 184-186:** Four rainfall events of total sampled events were chosen for detailed examine the relationship between DOC concentration ($C_i$) and flow rate in the hydrological process. These selected rainfall events represented 83% of the occurrence frequency of rainfall amount and the collected samples with high-frequency cover a complete of hydrological process during the monitoring period.

**Comment 4.** Please complete the name of the TOC analyser, like Vario TOC select or Vario TOC cube. I think it would be much better to describe what the 1% $H_3PO_4$ solution is used for.

**Response:** Thanks for your suggestions. These details show in Line 124-128 of this manuscript.

**Line 122-126:** DOC was recognized as the difference between total dissolved carbon (TDC) and dissolved inorganic carbon (DIC) for each sample (DOC=TDC-DIC). TDC and DIC were determined by Vario Select (Elementar, Germany), which included a high-temperature combustion furnace, a self-contained acidification module and a highly sensitive $CO_2$ detector. TDC was automatically measured by the combustion of a sample, whereas DIC was measured after acidified by 1% $H_3PO_4$ solution (phosphoric acid).

**Comment 5.** The meaning of DOC concentration and discharge should be consistent throughout the paper. DOC concentration has been defined as the flow-weighted mean DOC concentration. Ci and Qi were defined as the discharge and DOC in an individual runoff sample. However, the Y-axis in figure 5 corresponds to the discharge and DOC concentration of each sample. It is better to present flow-weighted mean DOC concentration in another way to distinguish it from the DOC in runoff samples and DOC in other studies.

**Response:** Thanks for your suggestions. In order to differentiate event-based DOC concentration and instant DOC concentration, these details have been changed in text and figures.

**Line 131-132:** In the present study, the flow-weight mean concentration ($C_f$) was used to determine the average DOC concentration in a rainfall event. $C_f$ was calculated by dividing the total DOC load by the total discharge in an event time.

**Line 136-137:** where, $Qi$ (L) is the discharge amount corresponding to sample $i$, which was calculated by flow rate and interval time; $C_i$ (mg L$^{-1}$) is the DOC concentration in a runoff sample $i$;

**$C_f$ in Y axis of Figure 4-b:**

[Figure]

**$C_i$ in Y axis of Figure 5 and Figure 6**

[Figure]

**Figure 6**

[Figure]

**Comment 6.** It would be better to use commas instead of semicolons. There should be a space between "7" and "and" (line 155-156). Do not use a colon to explain abbreviations (line 156).

**Response:** Thanks for your remind. The details have changed in Line 143-147:

**Line 143-147:** These variables are Q (total discharge volume a rainfall), Ra (total rainfall amount in a rainfall event), R1, R7 and R14 (total rainfall amount in the 1, 7 and 14 days before the current rainfall event, respectively), SMC-7 and SMC-14 (soil moisture content in the 7 and 14 days before the current rainfall event), $T_{air}$-7 and $T_{air}$-14 (mean air temperature in the 7 and 14 days before the current rainfall event) and REI (interval days between the current and last rainfall event).

**Comment 7.** Please use the version information of SPSS instead of Statistics Package for Social Science.

**Response:** Thanks for your remind. We have added the information in Line 149-151.

**Line149-151:** To analyze potential relationships among DOC concentration, flux, and selected variables, Pearson's test was performed using SPSS (Statistics Package for Social Science, Version 22).

**Comment 8.** I think the relationship between runoff and DOC concentration could be better explained by the DOC concentration of each runoff sample and its flow rate in the corresponding sampling period.

**Response:** Thanks for your remind. The discharge in this manuscript is flow rate in a hydrological process. We have changed in the Line 176-179 and the Figure 5 showed the DOC concentration of each runoff sample and its flow rate in the corresponding sampling period.

**Line 176-179:** The relationship between flow rate and $C_f$ for sampled rainfall events was shown in Figure 4-b. The $C_f$ exhibited a poor relationship with flow rate, and the $C_f$ was a more variable at low flow rate period compared to the high flow rate period, which is typically observed during consecutive rainfall events with high rainfall amount.

**Figure 5**

[Figure]

**Comment 9.** This part is the interpretation or analysis of the above results, and it should be included in the section of discussion with the citation.

**Response:** Thanks for your suggestions. This part has been rewritten and moved to discussion section Line 250-255.

**Line250-255:** The antecedent rainfall may increase connectivity in hydrology and DOC source contributed to runoff. Thus, the dilution effect diminished as flow rate decreased and the increased connectivity lead to a relatively higher DOC concentration during the falling limb (Hope et al., 1994;

Ma et al., 2018; Williams et al., 2017). A clockwise hysteresis was observed in 13-July and 10-September. The rapid response of flow rate to rainfall can be attributed to the rainfall event with a shorter duration and larger rainfall amount. The higher discharge may bring a higher flushing capacity, thus an increased DOC concentration was observed during the rising limb (Blaen et al., 2017; Tunaley et al., 2017).

**Comment 10.** What is the difference between DOC export and DOC flux? DOC export and flux were mixed in several parts of the paper affecting readers' understanding of the study. Please explain the specific meaning of DOC flux/export.

**Response:** Thanks for your suggestions. In this study, *Flux* emphasize the amount of DOC loaded by a rainfall event and DOC export mean a transport process occurs in a hydrological process from a catchment. Line 136-137 has added the specific meaning of *Flux*.

**Line 136-137:** *Flux* (kg km$^{-2}$) is the quantity of DOC driven by a rainfall event for in the study region; and, *s* is the catchment area (km$^2$).

**Comment 11.** Substitute "uses" with "used". As shown in figure 2-b, the DOC concentration looked like the average DOC concentration daily or per rainfall event, rather than the results of each collected runoff samples. This figure did not show the high-frequency monitoring in your study.

**Response:** Thanks for your suggestions. Figure 2-b showed the event-driven DOC concentration during the monitoring period. However, the high-frequency monitoring mean the runoff samples collected in a hydrological process in this manuscript, details showed in method part in Line 102-104:

**Line 102-104:** High-frequency monitoring was carried out in a rainfall event based hydrological process, thus the ISCO was set to acquire samples every 10 min from the first 12 runoff samples and another 12 were sampled every 30 min.

The "uses" revised to "used" in Line 226-228:

**Line 226-228:** In this study, we used an in-situ auto- and high-frequency monitoring method to observe temporal changes in hydrological and DOC concentration for an event-based sampling period during the concentrated rainfall season (June-September, 2016) (Figure 2-b).

**Comment 12.** Is the result of the greater DOC flux with a large discharge obtained from your study? If not, it is better not to compare the result about DOC flux in previous studies with the result of DOC concentration in your study.

**Response:** Thanks for your suggestions. In line 230-245, these words describe the monthly DOC flux showed no linear relationship with discharge amount and discussed the potential reason. In line 272-276, these words discussed the event-driven DOC flux during the monitoring period. Details

showed as following:

**Line 230-245:** Monthly DOC fluxes were not clearly correlated with discharge amount. The flow-weighted DOC concentrations decreased during the experimental period, which differed from the greater DOC flux with a large discharge (Chen et al., 2012; Cooper et al., 2007). Furthermore, the monthly DOC fluxes were negatively correlated with the discharge amount from June to August 2016. The DOC concentration was higher in June and decreased in August. This was reasonable because the accumulated soil organic carbon can be flushed by runoff in early rainfall period, and the DOC concentration may be diluted by increased runoff (Blaen et al., 2017; Chen et al., 2012). In addition, in combination with the increased discharge amount, the decreased concentration led to a decrease in monthly DOC flux from June to August. This could be explained by the relative changes in DOC concentrations being higher than changes in monthly discharge, indicating that the decreased concentration may outweigh the effect of increased discharge. However, the exception occurred in September, while increased DOC flux over the other three months was mainly due to a smaller increase in DOC concentration. These results were also probably associated with rainfall amount, land cover and runoff flow path (Laudon et al., 2004; Soulsby et al., 2003). For example, crops planted in the check-dam field were harvested, and the ratio of rainfall to runoff increased in September. The soil soluble organic carbon is more likely to leach through macropores from check-dam farmland into runoff, which further increased the DOC concentration in runoff. Thus, it led to a slight increase in DOC flux in September. Therefore, it could be inferred from these results that DOC flux may depend on runoff flushing capacity and flow path in a restored and check-dam catchment.

**Line 261-272:** For event-driven flux, the DOC flux is a function of total runoff discharge and DOC concentration ($C_f$). DOC flux showed a positive linear relationship with runoff discharges, which is not surprising and parallel with studies reported by Clark et al. (2007) and Ma et al. (2018). In addition, it should be noted that the DOC flux induced by larger rainfall amount was higher than flux driven by light rainfall, whereas the $C_f$ showed no evident difference for the selected rainfall events. Thus, the greater DOC flux clearly showed that the DOC export was close linked to hydrologic process induced by various amount of rainfall event in LPR.

**Comment 13.** line 266-267, line 274-277, line 300-301; and line 308-310;The above sentences are the description of the figures or tables and it is best to move those to the result section.

**Response:** Thanks for your suggestions. These sentences have been moved to the results part and details showed as following:

**Line 168-169:** In addition, Figure 4-a showed the relationship between flow rate and rainfall amount during June to September.

**Line 169-170:** This indicated that event-driven flow rate varied with rainfall amount, and thus suggested that runoff discharges are highly sensitive to larger rainfall amount with greater than 20 mm in this area.

**Line 179-182:** Table 2 showed the correlation between $C_f$ and a set of factors in all sampled rainfall events during the study period. On one hand, the $C_f$ was positively correlated with rainfall amount (Ra) and R7. On the other hand, the $C_f$ was extreme significantly and negatively correlated with SMC7 and SMC14.

**Line 214-217:** The relationship between event-based DOC flux and runoff discharge amount is shown in Figure 4-c. The DOC flux showed a positive linear relationship with the runoff discharge amount, especially for violent rainfall events. The DOC flux was more variable in lower runoff discharge conditions. In general, event-based DOC flux was significantly and positively correlated with Q, Ra, R1 and R, as showed in Table 2.

**Comment 14.** Why do you take the 20 mm rainfall amount as the break point to do the linear regression analysis respectively?

**Response:** Thanks for your suggestions. We have added a finding conducted by Yang and details show in the following:

**Line 156-159:** All the rainfall events in between June to September were grouped into four grades: <5 mm (Light rainfall), 5-10 mm (Moderate rainfall), 10-20 mm (Heavy rainfall), and >20 mm (Violent rainfall) according to rainfall amount classification (Yang et al., 2018).

**Comment 15.** line 288; The subtitle is too broad and general. Rainfall, one of the most important factors affecting DOC concentration, has been mentioned in 4.1, but not been fully discussed.

**Response:** Thanks for your suggestions. The subtitle has changed into "4.2 Potential Factors Influence on DOC Export". We have added some information in this discussion part and details show in the following:

**Line282-314:** The infrequent and amount of violent rainfall events strongly influence the runoff discharges and soil moisture, which in turn impact on DOC during or later export from a catchment. In this study, temporal variations of rainfall, air temperature and soil moisture content were continuously monitored throughout the study period to provide detailed information describing the antecedent and current conditions. Positively correlation between Ra, R7 and $C_f$ suggested that the combination of the current rainfall amount and the accumulated rainfall before a current rainfall event are important. R7 may reflect the antecedent hydrological condition and Ra represent the current rainfall input into the catchment, resulting in well hydrological connectivity, and more DOC source may contribute to runoff. Therefore, $C_f$ can by strongly influenced by Ra and R7 due to the hydrological properties of the catchment. Apart from the hydrological changes, the antecedent soil moisture also played an important role in $C_f$ and showed an extreme significantly and negatively correlated with SMC7 and SMC14 (Table 2). The soil moisture content was continuously dried and then effectively rewetted under a specific rainfall amount, as supported by the soil moisture variations shown in Figure 2-c. These results were also consistent with Yang et al. (2018), who found that the threshold of rainfall effectively recharged

into soil was 20-26 mm for grassland and forestland in LPR. Therefore, the pattern of soil moisture dry-wet cycle may affect event-driven DOC concentration, and this highlights the importance of soil moisture condition in DOC export (Figure 7). The higher DOC concentrations from June to middle July coincided with light rainfall, and thus rainfall recharge into soil moisture. This is probably attributed to inactive microbial activity, caused by the relatively lower soil moisture (Jager et al., 2009). The DOC concentration decreased with increased soil moisture content, particularly in July-18 with a total rainfall amount of 56.4 mm. On one hand, violent rainfall events may induce a higher discharge, causing a dilution effects on DOC concentration. On the other hand, the rainfall water may effectively replenish soil moisture content, and thus stimulate a higher decomposition of soil carbon under wet and higher temperature condition. Then, the relative decreased DOC concentrations were observed in a drying soil moisture condition for the next rainfall events, which may attribute to an exhaustion of DOC (Laudon et al., 2004). These findings were similar to previous studies by Tunaley et al. (2017), who reported a strong influence of dry antecedent conditions on DOC export response to rainfall event.

For event-based flux, DOC flux was significantly and positively correlated with Q, Ra, R1 and R7. The Q and Ra reflect the direct effect of current rainfall and hydrological processes during a rainfall event, while R1 and R7 refer to the antecedent rainfall conditions and reflect indirect effects on DOC export. These results agreed with previous studies demonstrated by Blaen et al. (2017), who noted that antecedent conditions and rainfall were key drivers of DOC export during a rainfall event. Cooper et al. (2007) also concluded that DOC export is largely governed by interactions between hydrological and meteorological factors and carbon biogeochemical process. Overall, these results suggested that rainfall is a key factor influencing hydrological process, and thus DOC export from an ecological restored catchment in LPR. Apart from the increased soil carbon driven by increased vegetation (Wang et al., 2011b), the weaken hydrological process induced by increased vegetation may also cause a less terrestrial carbon export from a catchment. Therefore, our results highlight the need for research not only into the hydrological process and soil carbon cycle, but the integration of carbon export driven by a sequence of rainfall events across spatiotemporal scales to understand the carbon balance in a restored catchment in LPR.

**Comment 16.** Figure 2; Try to use shading or background fill to distinguish the values of sampling days instead of using different colored dots.

**Response:** Thanks for your suggestions. The Figure 2 has been revised as following:

[Figure]

**Comment 17.** Figure 4; Is the regression curve in the right side of figure 4 fitted according to all sampled rainfall events or according to >20 mm rainfall events? Please indicate (a) and (b) in the figure 4.

**Response:** Thanks for your suggestions. For the Figure 4-c, we removed the relationship between DOC flux and discharge amount due to the flux was calculated by discharge amount. Thus, the Figure 4 has been changed and added (a), (b), (c), respectively. The details shown in Figure 4 as following:

**Figure 4**

[Figure]

**Other Changes:**

We have revised the abstract part in Line 11-26 and the conclusion part in Line 324-331. We also added discussion information in Line 261-271 and Line 304-314. The details show in the following part:

In Abstract **Line 11-26:** Dissolved organic carbon (DOC) transported by runoff has been identified as an important role of the global carbon cycle. Despite there being many studies on DOC concentration and flux, but little information is available in semi-arid catchments of the Loess Plateau Region (LPR). The primary goal of this study was to quantify DOC exported driven by a sequence of rainfall events during the concentrated rainfall season. In addition, factors that affect DOC export from a small headwater catchment will be investigated accordingly. Runoff discharge and DOC concentration were monitored at the outlet of the Yangjuangou catchment in Yanan, Shaanxi Province, China. The results showed that DOC concentration was highly variable, with event-based DOC concentrations ranging from 4.08 to 15.66 mg L$^{-1}$. Hysteresis analysis showed a nonlinear relationship between DOC concentration and flow rate in the hydrological process. The monthly DOC flux loading from the catchment was 94.73-110.17 kg km$^{-2}$ from June to September, while the event-based DOC flux ranged from 0.18 to 2.84 kg km$^{-2}$. Variations of event-driven DOC concentration contributed slightly to a difference in DOC flux, whereas intra-events of rainfall amount and runoff discharge led to evident difference in DOC export. In conclusion, our case results highlighted the advantages of high-frequency

monitoring for DOC export and indicated that event-driven DOC export is largely influenced by the interaction of catchment hydrology and antecedent condition within a catchment. Engineering and scientists can take advantage of the derived results to better develop advanced field monitoring work. In addition, more studies are needed to investigate the magnitude of terrestrial DOC export in response to projected climate change at larger spatiotemporal scale, which may have implication for the carbon balance and carbon cycle model from an ecological restored catchment in LPR.

**Line 261 -271:** For event-driven flux, the DOC flux is a function of total runoff discharge and DOC concentration ($C_f$). DOC flux showed a positive linear relationship with runoff discharges, which is not surprising and parallel with studies reported by Clark et al. (2007) and Ma et al. (2018). In addition, it should be noted that the DOC flux induced by larger rainfall amount was higher than flux driven by light rainfall, whereas the $C_f$ showed no evident difference for the selected rainfall events. Thus, the greater DOC flux clearly showed that the DOC export was close linked to hydrologic process induced by various amount of rainfall event in LPR. For an ecological restored catchment in LPR, the soil carbon driven by increased vegetation was significantly increased and acted as a positive pathway to sequestration soil carbon on terrestrial ecosystem (Wang et al., 2011b). Meanwhile, the reduced hydrology responded to an increased vegetation may diminish soil carbon transported by hydrological process in a catchment. The event-driven DOC transport is an important component for evaluating carbon balance of the ecological restored catchment in LPR. Hence, further study should be long-term undertaking to investigate the hydrological response and its impact on terrestrial carbon loss from a catchment in LPR.

**Line 304-314:** DOC flux was significantly and positively correlated with Q, Ra, R1 and R7. The Q and Ra reflect the direct effect of current rainfall and hydrological processes during a rainfall event, while R1 and R7 refer to the antecedent rainfall conditions and reflect indirect effects on DOC export. These results agreed with previous studies demonstrated by Blaen et al. (2017), who noted that antecedent conditions and rainfall were key drivers of DOC export during a rainfall event. Cooper et al. (2007) also concluded that DOC export is largely governed by interactions between hydrological and meteorological factors and carbon biogeochemical process. Overall, these results suggested that rainfall is a key factor influencing hydrological process, and thus DOC export from an ecological restored catchment in LPR. Apart from the increased soil carbon driven by increased vegetation (Wang et al., 2011b), the weaken hydrological process induced by increased vegetation may also cause a less terrestrial carbon export from a catchment. Therefore, our results highlight the need for research not only into the hydrological process and soil carbon cycle, but the integration of carbon export driven by a sequence of rainfall events across spatiotemporal scales to understand the carbon balance in a restored catchment in LPR.

**Line 324-331:** These results showed that the temporal variation magnitude of DOC is related to hydrological condition and antecedent condition, and suggested that the event-driven DOC export is largely influenced by rainfall through direct effects on catchment hydrology and indirect effects on soil carbon cycles. Changes in catchment hydrology and soil carbon processes responded to climate change may play an important role in terrestrial carbon export, in particular for a restored catchment. Thus,

further work should focus on carbon export response to various rainfall events at a larger spatiotemporal scale for better estimating future terrestrial carbon flux to aquatic ecosystem and evaluating carbon balance in ecological restored catchment in LPR. In addition, engineers and scientists can take advantage of the derived results to better develop advanced field monitoring work.

---

## Author Comment (AC2) · 11 Apr 2019

**Response to HESS-2019-8-RC2**

**Anonymous Referee #2**

Thanks for your suggestions. We appreciate for anonymous referee comments concerning our manuscript entitled "Dissolved Organic Carbon Driven by Rainfall Events from a Semi-arid Catchment during Concentrated Rainfall Season in the Loess Plateau, China". We have studied comments carefully and have made corrections. The main corrections in the manuscript according to the referee's comments are as follows:

**Comment 1.** L43-50: these piled data didn't give a clear background on DOC export. They should be re-organized and present in term of different catchment characteristics.

**Response:** Thanks for your suggestions. These sentences of the introduction have been re-written in Line 40-59 as detail shown as following:

**Line 40-57:** DOC exported from catchments has attracted great attention in the last two decades due to global concerns about potential influences on the global carbon cycle and climate change (Laudon et al., 2004; Raymond et al., 2013). The transport of terrestrial DOC to runoff is strongly influenced by hydrological process, soil carbon cycle and climatological factors. Hydrological process driven by rainfall event plays an important role in controlling terrestrial DOC from soil pool to runoff. Previous studies have shown that the release of DOC concentrations ranged from 0.5 to 50 mg $L^{-1}$ for global catchments (Mulholland, 2003). For instance, Clark et al. (2007) found that DOC concentration ranged between 5-35 mg $L^{-1}$ with a highly variable in rainfall events from a peatland catchment, and a study by Blaen et al. (2017) showed that the DOC concentration ranged from 5.4 to 18.9 mg $L^{-1}$. Similar results were reported by Ran et al. (2018), who found that DOC concentration ranged from 1.4 to 9.5 mg $L^{-1}$ in the Wuding River in the LPR. Such studies highlighted that the importance of hydrological process on DOC transport (Billett et al., 2006; Dawson et al., 2002; Inamdar et al., 2006). Different rainfall events may alter hydrological connectivity or the flow path, which in turn lead to a varied hydrological connectivity and DOC source contributing to runoff. Moreover, the intensity and frequency of rainfall event not only influenced the current hydrological and DOC loading processes, but also changed the soil moisture conditions. The latter point may be particularly important in soil biogeochemical cycle. For example, DOC concentration may increase due to accumulated soil organic carbon after a dry period (Jager et al., 2009). In addition, variations in the magnitude and frequency of precipitation are one of manifestations of climate change, and thus, changes in hydrological process induced by climate change are also impact on the transport of terrestrial DOC. Therefore, understanding the dynamic and magnitude of DOC export from catchment is an important component of prediction DOC flux under the circumstance of future climate change.

**Comment 2.** The knowledge gap is not well stated.

**Response:** Thanks for your suggestions. The knowledge gap and objective part has been reorganized as following:

**Line 69-79:** Less information is available on DOC export driven by rainfall event, which DOC flux is an important component in overall carbon balance for ecological restored catchment.

Therefore, the primary goal of this study is to investigate how variations of DOC concentration and flux response to a sequence of rainfall events from a restored catchment during concentrated rainfall season in the LPR. Specifically, the two objectives of this study were (1) to examine the dynamic changes in DOC concentration and flux and assess the difference in DOC export driven by various rainfall events, and (2) evaluate how rainfall, runoff, and antecedent factors affect DOC export from a catchment. To do so, we used high-frequency method to capture the temporal changes in DOC export and hydrological process driven by rainfall event within an ecological restored watershed in LPR. These results will provide evidence of DOC export response to rainfall events, especially driven by extreme events, which may be important for evaluating carbon balance and modeling DOC export through runoff at ecological restored catchment in LPR.

**Comment 3.** Be more specific about the experiment duration. How long/how many rainfall events have you been monitoring and sampling?

**Response:** Thanks for your suggestions. Line 108-109 added the sampling information.

**Line 108-109:** There were 278 samples collected for 22 hydrological processes induced by rainfall event over the monitoring period of June to September, 2016.

**Comment 4.** The author should either consider combining the result and discussion sections OR separating them clearly in the writing. There are multiple places that the results been re-stated in discussion or discussed the result right after without citation.

**Response:** Thanks for your suggestions. Some sentences in discussion part has been moved to the results part and details showed as following:

**Line 168-169:** In addition, Figure 4-a showed the relationship between flow rate and rainfall amount during June to September.

**Line 169-170:** This indicated that event-driven flow rate varied with rainfall amount, and thus suggested that runoff discharges are highly sensitive to larger rainfall amount with greater than 20 mm in this area.

**Line 179-182:** Table 2 showed the correlation between $C_f$ and a set of factors in all sampled rainfall events during the study period. On one hand, the $C_f$ was positively correlated with rainfall amount (Ra) and R7. On the other hand, the $C_f$ was extreme significantly and negatively correlated with

SMC7 and SMC14.

**Line 214-217:** The relationship between event-based DOC flux and runoff discharge amount is shown in Figure 4-c. The DOC flux showed a positive linear relationship with the runoff discharge amount, especially for violent rainfall events. The DOC flux was more variable in lower runoff discharge conditions. In general, event-based DOC flux was significantly and positively correlated with Q, Ra, R1 and R, as showed in Table 2.

**Comment 5.** I'm confused with the way you separate the rainfall events into 4 groups. In Figure 3 you stated in x-axis was rainfall intensity, but the unit was mm, not mm/h. Why do you define rainfall intensity based on accumulated rainfall depth? In the Yang et al. (2018) paper you referred, they denoted the rainfall replenishment in mm to effectively recharge the soil water.

**Response:** Thanks for your suggestions. Indeed, the rainfall events were grouped by rainfall amount and the Figure 3 has been changed. According to Yang's results, the threshold of rainfall amount mean rainwater can effectively recharge the soil water in LPR, which may affect soil moisture content. This is why we selected this classification. Thus, we choose the parameter of rainfall amount to analyze in this manuscript.

**Line 156-158:** All the rainfall events in between June to September were grouped into four grades: <5 mm (Light rainfall), 5-10 mm (Moderate rainfall), 10-20 mm (Heavy rainfall), and >20 mm (Violent rainfall) according to rainfall amount classification (Yang et al., 2018).

**Figure 3:**

[Figure]

**Line 246-260:** Despite the facts that the DOC export varied in different months, there were also differences in DOC concentration and flux response to a rainfall event. DOC concentrations exhibited different dynamic changes throughout an event-driven hydrological process. In our result, the anticlockwise hysteresis between DOC concentration and flow rate was observed at 6-June. The peak DOC concentration was delayed compare to peak flow rate. These results may be attributed to a 5.2 mm

rainfall was happen earlier than the maximum rainfall at 6-June (Figure 5-a). The antecedent rainfall may increase connectivity in hydrology and DOC source contributed to runoff. Thus, the dilution effect diminished as flow rate decreased and the increased connectivity lead to a relatively higher DOC concentration during the falling limb (Hope et al., 1994; Ma et al., 2018; Williams et al., 2017). A clockwise hysteresis was observed in 13-July and 10-September. The rapid response of flow rate to rainfall can be attributed to the rainfall event with a shorter duration and larger rainfall amount. The higher discharge may bring a higher flushing capacity, thus an increased DOC concentration was observed during the rising limb (Blaen et al., 2017; Tunaley et al., 2017). Moreover, the close link of DOC source to runoff may lead to a rapid increased in DOC concentration. A figure-of-eight hysteresis was observed in 2-August. due to the DOC concentration keep pace with flow rate during the rising and falling limb. Moreover, the event-driven DOC concentration at 2-August showed no distinct difference with other three higher rainfall amount events. These results suggested that a lower discharge induced by lower rainfall amount have a more complex and larger influence on DOC concentration from a catchment in LPR.

**Comment 6.** The major finding you stated was higher DOC export with low DOC concentration. I have several questions about this finding: In Figure 4, you stated DOC concentration depressed with increased discharge for greater intensity.

**Response:** Thanks for your suggestions. The conclusion has been rewritten as following:

**Line 321-324:** These results showed that higher DOC flux with low DOC concentration related to higher discharge and its dilution effects in a hydrological process driven by larger rainfall amount. The diluted DOC concentration induced by increased discharges contributed slightly to difference in DOC flux, due to total runoff discharge is a major variable for flux.

**Comment 7.** How are you sure since you only have 5 points with r2 value of 0.38. Is this correlation significant? In Figure 5, DOC do show positive relationship with discharge within individual event, how do you explain this contrary?

**Response:** Thanks for your suggestions. The regression has been removed. The results shown in Figure 5 has been reorganized in discussion part:

[Figure]

**Comment 8.** L32: insert a summary sentence before "For instance, high DOC. . .". The following statements come from nowhere and it's confusing.

**Response:** Thanks for your suggestions. The first paragraph in introduction has been reorganized and the details show in Line 28-39 of this manuscript:

**Line 28-39:** Dissolved organic carbon (DOC), often defined as the solute filtered through <0.45μm pore size, is regarded as one of the active constituents and provides a biologically available carbon source for organisms (Raymond and Saiers, 2010). The estimated DOC flux of terrestrial organic carbon through major worldwide rivers to ocean is from 0.45 to 0.78 Pg C y⁻¹(Drake et al., 2018; Hedge et al., 1997; Ran et al., 2018). The substantial magnitude of flux suggests that the DOC export on a global scale acts as one of the crucial processes of linking between terrestrial and aquatic ecosystem (Battin et al., 2008; Raymond et al., 2013; Raymond and Saiers, 2010). For instance, high DOC concentrations can lead to water pollution and eutrophication, and thus have dramatic consequences on aquatic ecosystem services (Evans et al., 2005; Hu et al., 2016). In addition to ecological impacts, DOC in runoff also play an important role in social well-beings. High DOC concentrations will aggravate the complexation and adsorption of pesticides and heavy metals in hydrological process. Therefore, the quality of domestic water could be damaged and it might potentially lead to adverse impacts on human health, such as increased risk of cancer, diabetes, or other diseases (Bennett et al., 2009; Ritson et al., 2014). Therefore, it is urgent to improve the associated knowledge on DOC export variability and develop a mechanistic understanding of DOC export from catchments.

**Comment 9.** give the time period for average annual temperature and precipitation. Is 535 mm only coming from rainfall or also including other type of precipitation?

**Response:** Thanks for your suggestions. We have added some information about the precipitation and details show in the following:

Line 86-88: The climate of this catchment is situated in a semi-arid continental monsoonal climate with an average annual temperature of 9.6℃ and average annual precipitation is 535 mm during the period from 1951 to 2012 (Li and Wang, 2015).

**Comment 10.** state specific land alteration in "represent an area with altered land use that has. . ."

**Response:** Thanks for your suggestions. These details show in Line 92-94 of this manuscript

**Line 92-94:** The proportion of sloping cropland has remarkably decreased from 16.9% in 1998 to 0.1% in 2006. The forestland increased from 15.2% in 1998 to 37.4% in 2006 since implemented the 'Grain-for-Green' and engineering measures (Wang et al., 2011b).

**Comment 11.** the part "In addition, the aim of hydrological. . ." should be stated before you introduce the meteorological station, and also should be condensed.

**Response:** Thanks for your suggestions. The sentence has been moved forward to Line 109-110.

**Line 109-110:** In addition, the aim of hydrological and meteorological factor monitoring was to characterize the temporal changes of catchment condition.

**Comment 12.** L133: "microbiologically biodegrade" to "microbially degrade".

**Response:** Thanks for your suggestions. The "microbiologically biodegrade" has been changed to "microbially degrade" in Line 118-119.

**Line 118-119:** In the Yangjuangou catchment, researchers resided in the field observatory station and treated the samples immediately after a rainfall event to ensure that the DOC in the sampled water did not microbially degrade.

**Comment 13.** L138-L140: give CV of procedure accuracy.

**Response:** Thanks for your suggestions. We added the CV of procedure accuracy in Line 127-128:

Line 127-128: In order to control quality, each sample is determined through analysis of two replicate and the coefficient of variation of tested results was less than 10%.

**Comment 14.** L149-L156: this section should be in laboratory analysis or an independent section rather in data analysis.

**Response:** Thanks for your suggestions. The section has been reorganized

**Line 129-152:**

**2.4 Data Analysis**

**2.4.1 Event-driven DOC Concentration and Flux Calculation**

**2.4.2 Variables related to Event-driven DOC Transport**

**2.4.3 Statistical analysis**

**Comment 15.** L166: "in June to September" to "in between June and September".

**Response:** Thanks for your remind.

**Line 156-158:** All the rainfall events in between June to September were grouped into four grades: <5 mm (Light rainfall), 5-10 mm (Moderate rainfall), 10-20 mm (Heavy rainfall), and >20 mm (Violent rainfall) according to rainfall amount classification (Yang et al., 2018).

**Comment 16.** L186: I suggest to open this paragraph with sentence "In general, runoff discharge tended to follow the pattern of rainfall amount in the study catchment."

**Response:** Thanks for your suggestions. we changed at the beginning of this paragraph in Line 163:

**Line 163:** In general, runoff discharge tended to follow the pattern of rainfall amount in the study catchment.

**Comment 17.** L189: where did the value "34.70 mg L$^{-1}$" come from? I didn't see this value in Figure 2 or Figure 5.

**Response:** Thanks for your remind. The value has been changed in Line 175-176:

**Line 175-176:** For the event-driven DOC concentration, the flow-weight mean DOC concentration ($C_f$) ranged from 4.08 to 15.66 mg L$^{-1}$ for all sampled rainfall events during June to September.

**Comment 18.** L191: "DOC concentration were less variable during June to September", less compare to what?

**Response:** Thanks for your suggestions. We revised this sentence in Line 185-186:

**Line 174-175:** There were less variations in the mean DOC concentration among monitoring months.

**Comment 19.** Figure 3: see previous comments about the grouping.

**Response:** Thanks for your suggestions. The rainfall events were grouped by rainfall amount and the Figure 3 has been changed.

[Figure]

**Comment 20.** Figure 4: Is the second figure necessary? DOC flux is calculated based on discharge. Why present a variable that is highly dependent on the other variable?

**Response:** Thanks for your suggestions. hanks for your suggestions. For the Figure 4-c, we removed the relationship between DOC flux and discharge amount due to the flux was calculated by discharge amount. Thus, the regression has been removed and the details shown as following:

**Figure 4**

[Figure]

**Comment 21.** Figure 5: explain in result section why did you choose these four events? Do they show different rainfall intensity? Axis of DOC concentration could be in the same scale.

**Response:** Thanks for your suggestions. Line 184-186 has been added and explained why we

choose these four events. The axis of DOC concentration in Figure 5 has been change to the same scale.

**Line 184-186:** Four rainfall events of total sampled events were chosen for detailed examine the relationship between DOC concentration ($C_i$) and flow rate in the hydrological process. These selected rainfall events represented 83% of the occurrence frequency of rainfall amount and the collected samples with high-frequency cover a complete of hydrological process during the monitoring period.

**Figure 5**

[Figure]

**Comment 22.** Figure 6: Axis of DOC concentration could be in the same scale.

**Response:** Thanks for your suggestions.

**Figure 6**

[Figure]

**Other Changes:**

We have revised the abstract part in Line 11-26 and the conclusion part in Line 324-331. We also added discussion information in Line 261-271 and Line 304-314. The details show in the following part:

In Abstract **Line 11-26:** Dissolved organic carbon (DOC) transported by runoff has been identified as an important role of the global carbon cycle. Despite there being many studies on DOC concentration and flux, but little information is available in semi-arid catchments of the Loess Plateau Region (LPR). The primary goal of this study was to quantify DOC exported driven by a sequence of rainfall events during the concentrated rainfall season. In addition, factors that affect DOC export from a small headwater catchment will be investigated accordingly. Runoff discharge and DOC concentration were monitored at the outlet of the Yangjuangou catchment in Yanan, Shaanxi Province, China. The results showed that DOC concentration was highly variable, with event-based DOC concentrations ranging from 4.08 to 15.66 mg L$^{-1}$. Hysteresis analysis showed a nonlinear relationship between DOC

concentration and flow rate in the hydrological process. The monthly DOC flux loading from the catchment was 94.73-110.17 kg km$^{-2}$ from June to September, while the event-based DOC flux ranged from 0.18 to 2.84 kg km$^{-2}$. Variations of event-driven DOC concentration contributed slightly to a difference in DOC flux, whereas intra-events of rainfall amount and runoff discharge led to evident difference in DOC export. In conclusion, our case results highlighted the advantages of high-frequency monitoring for DOC export and indicated that event-driven DOC export is largely influenced by the interaction of catchment hydrology and antecedent condition within a catchment. Engineering and scientists can take advantage of the derived results to better develop advanced field monitoring work. In addition, more studies are needed to investigate the magnitude of terrestrial DOC export in response to projected climate change at larger spatiotemporal scale, which may have implication for the carbon balance and carbon cycle model from an ecological restored catchment in LPR.

**Line 261 -271:** For event-driven flux, the DOC flux is a function of total runoff discharge and DOC concentration ($C_f$). DOC flux showed a positive linear relationship with runoff discharges, which is not surprising and parallel with studies reported by Clark et al. (2007) and Ma et al. (2018). In addition, it should be noted that the DOC flux induced by larger rainfall amount was higher than flux driven by light rainfall, whereas the $C_f$ showed no evident difference for the selected rainfall events. Thus, the greater DOC flux clearly showed that the DOC export was close linked to hydrologic process induced by various amount of rainfall event in LPR. For an ecological restored catchment in LPR, the soil carbon driven by increased vegetation was significantly increased and acted as a positive pathway to sequestration soil carbon on terrestrial ecosystem (Wang et al., 2011b). Meanwhile, the reduced hydrology responded to an increased vegetation may diminish soil carbon transported by hydrological process in a catchment. The event-driven DOC transport is an important component for evaluating carbon balance of the ecological restored catchment in LPR. Hence, further study should be long-term undertaking to investigate the hydrological response and its impact on terrestrial carbon loss from a catchment in LPR.

**Line 304-314:** DOC flux was significantly and positively correlated with Q, Ra, R1 and R7. The Q and Ra reflect the direct effect of current rainfall and hydrological processes during a rainfall event, while R1 and R7 refer to the antecedent rainfall conditions and reflect indirect effects on DOC export. These results agreed with previous studies demonstrated by Blaen et al. (2017), who noted that antecedent conditions and rainfall were key drivers of DOC export during a rainfall event. Cooper et al. (2007) also concluded that DOC export is largely governed by interactions between hydrological and meteorological factors and carbon biogeochemical process. Overall, these results suggested that rainfall is a key factor influencing hydrological process, and thus DOC export from an ecological restored catchment in LPR. Apart from the increased soil carbon driven by increased vegetation (Wang et al., 2011b), the weaken hydrological process induced by increased vegetation may also cause a less terrestrial carbon export from a catchment. Therefore, our results highlight the need for research not only into the hydrological process and soil carbon cycle, but the integration of carbon export driven by a sequence of rainfall events across spatiotemporal scales to understand the carbon balance in a restored catchment in LPR.

**Line 324-331:** These results showed that the temporal variation magnitude of DOC is related to hydrological condition and antecedent condition, and suggested that the event-driven DOC export is largely influenced by rainfall through direct effects on catchment hydrology and indirect effects on soil carbon cycles. Changes in catchment hydrology and soil carbon processes responded to climate change may play an important role in terrestrial carbon export, in particular for a restored catchment. Thus, further work should focus on carbon export response to various rainfall events at a larger spatiotemporal scale for better estimating future terrestrial carbon flux to aquatic ecosystem and evaluating carbon balance in ecological restored catchment in LPR. In addition, engineers and scientists can take advantage of the derived results to better develop advanced field monitoring work.

---

## Author Comment (AC3) · 11 Apr 2019

**Response to HESS-2019-8-RC3**

**Anonymous Referee #3**

Thanks for your suggestions. We appreciate for anonymous referee comments concerning our manuscript entitled "Dissolved Organic Carbon Driven by Rainfall Events from a Semi-arid Catchment during Concentrated Rainfall Season in the Loess Plateau, China". We have studied comments carefully and have made corrections. The main corrections in the manuscript according to the referee's comments are as follows:

**Comment 1.** The authors declared that there were no reports to the LPR, but I do not think that is the reason they conducted such a study.

**Response:** Thanks for your suggestions. The Line 72-79 has been revised and explained why we conducted this study.

**Line 72-79:** Therefore, the primary goal of this study is to investigate how variations of DOC concentration and flux response to a sequence of rainfall events from a restored catchment during concentrated rainfall season in the LPR. Specifically, the two objectives of this study were (1) to examine the dynamic changes in DOC concentration and flux and assess the difference in DOC export driven by various rainfall events, and (2) evaluate how rainfall, runoff, and antecedent factors affect DOC export from a catchment. To do so, we used high-frequency method to capturing the temporal changes in DOC export and hydrological process driven by rainfall event within an ecological restored watershed in LPR. These results will provide evidence of DOC export response to rainfall events, especially driven by extreme events, which may be important for evaluating carbon balance and modeling DOC export through runoff at ecological restored catchment in LPR.

**Comment 2.** They claimed that this study highlighted the interaction of rainfall and antecedent conditions for DOC exports in a catchment, but they did not say what interactions and what effects.

**Response:** Thanks for your suggestions. The Line 282-303 explained the antecedent condition for DOC exports.

**Line 282-303:** The infrequent and amount of violent rainfall events strongly influence the runoff discharges and soil moisture, which in turn impact on DOC during or later export from a catchment. In this study, temporal variations of rainfall, air temperature and soil moisture content were continuously monitored throughout the study period to provide detailed information describing the antecedent and current conditions. Positively correlation between Ra, R7 and $C_f$ suggested that the combination of the current rainfall amount and the accumulated rainfall before a current rainfall event are important. R7 may reflect the antecedent hydrological condition and Ra represent the current rainfall input into the catchment, resulting in well hydrological connectivity, and more DOC source may contribute to runoff.

Therefore, $C_f$ can by strongly influenced by Ra and R7 due to the hydrological properties of the catchment. Apart from the hydrological changes, the antecedent soil moisture also played an important role in $C_f$ and showed a extreme significantly and negatively correlated with SMC7 and SMC14 (Table 2). The soil moisture content was continuously dried and then effectively rewetted under a specific rainfall amount, as supported by the soil moisture variations shown in Figure 2-c. These results were also consistent with Yang et al. (2018), who found that the threshold of rainfall effectively recharged into soil was 20-26 mm for grassland and forestland in LPR. Therefore, the pattern of soil moisture dry-wet cycle may affect event-driven DOC concentration, and this highlights the importance of soil moisture condition in DOC export (Figure 7). The higher DOC concentrations from June to middle July coincided with light rainfall, and thus rainfall recharge into soil moisture. This is probably attributed to inactive microbial activity, caused by the relatively lower soil moisture (Jager et al., 2009). The DOC concentration decreased with increased soil moisture content, particularly in July-18 with a total rainfall amount of 56.4 mm. On one hand, violent rainfall events may induce a higher discharge, causing a dilution effects on DOC concentration. On the other hand, the rainfall water may effectively replenish soil moisture content, and thus stimulate a higher decomposition of soil carbon under wet and higher temperature condition. Then, the relative decreased DOC concentrations were observed in a drying soil moisture condition for the next rainfall events, which may attribute to an exhaustion of DOC (Laudon et al., 2004). These findings were similar to previous studies by Tunaley et al. (2017), who reported a strong influence of dry antecedent conditions on DOC export response to rainfall event.

**Comment 3.** The introduction is very difficult to follow, they presented a numerous report (for example, L40-L64), I think they need to summary these studies and then the potential readers can know why they design this study.

**Response:** Thanks for your suggestions. The introduction part has been rewritten and the details show in Line 25-79 of this manuscript:

**Line 28-79:** Dissolved organic carbon (DOC), often defined as the solute filtered through <0.45μm pore size, is regarded as one of the active constituents and provides a biologically available carbon source for organisms (Raymond and Saiers, 2010). The estimated DOC flux of terrestrial organic carbon through major worldwide rivers to ocean is from 0.45 to 0.78 Pg C y$^{-1}$(Drake et al., 2018; Hedge et al., 1997; Ran et al., 2018). The substantial magnitude of flux suggests that the DOC export on a global scale acts as one of the crucial processes of linking between terrestrial and aquatic ecosystem (Battin et al., 2008; Raymond et al., 2013; Raymond and Saiers, 2010). For instance, high DOC concentrations can lead to water pollution and eutrophication, and thus have dramatic consequences on aquatic ecosystem services (Evans et al., 2005; Hu et al., 2016). In addition to ecological impacts, DOC in runoff also play an important role in social well-beings. High DOC concentrations will aggravate the complexation and adsorption of pesticides and heavy metals in hydrological process. Therefore, the quality of domestic water could be damaged and it might potentially lead to adverse impacts on human health, such as increased risk of cancer, diabetes, or other diseases (Bennett et al., 2009; Ritson et al., 2014). Therefore, it is urgent to improve the associated knowledge on DOC export variability and

develop a mechanistic understanding of DOC export from catchments.

DOC exported from catchments has attracted great attention in the last two decades due to global concerns about potential influences on the global carbon cycle and climate change (Laudon et al., 2004; Raymond et al., 2013). The transport of terrestrial DOC to runoff is strongly influenced by hydrological process, soil carbon cycle and climatological factors. Hydrological process driven by rainfall event plays an important role in controlling terrestrial DOC from soil pool to runoff. Previous studies have shown that the release of DOC concentrations ranged from 0.5 to 50 mg L$^{-1}$ for global catchments (Mulholland, 2003). For instance, Clark et al. (2007) found that DOC concentration ranged between 5-35 mg L-1 with a highly variable in rainfall events from a peatland catchment, and a study by Blaen et al. (2017) showed that the DOC concentration ranged from 5.4 to 18.9 mg L-1. Similar results were reported by Ran et al. (2018), who found that DOC concentration ranged from 1.4 to 9.5 mg L-1 in the Wuding River in the LPR. Such studies highlighted that the importance of hydrological process on DOC transport (Billett et al., 2006; Dawson et al., 2002; Inamdar et al., 2006). Different rainfall events may alter hydrological connectivity or the flow path, which in turn lead to a varied hydrological connectivity and DOC source contributing to runoff. Moreover, the intensity and frequency of rainfall event not only influenced the current hydrological and DOC loading processes, but also changed the soil moisture conditions. The latter point may be particularly important in soil biogeochemical cycle. For example, DOC concentration may increase due to accumulated soil organic carbon after a dry period (Jager et al., 2009). In addition, variations in the magnitude and frequency of precipitation are one of manifestations of climate change, and thus, changes in hydrological process induced by climate change are also impact on the transport of terrestrial DOC. Therefore, understanding the dynamic and magnitude of DOC export from catchment is an important component of prediction DOC flux under the circumstance of future climate change.

The LPR, which has an area of $6.4 \times 10^5$ km$^2$, is situated in the middle reaches of the Yellow River, China, and approximately 90% of the river loading sediment is derived from this region (Tang, 2004). With regard to this fragile environment, the Chinese government has launched some ecological restoration projects since the beginning of this century, such as the 'Grain-for-Green' and 'Natural Forest Protection Project'. With the implementation of these projects, large areas of steep-sloping (higher than 20°) agricultural land was converted to forest, shrub, or grassland, and engineering measures were also applied to control erosion (Fu et al., 2017). For instance, check dams can retain sediment and also offer flat and fertile land behind the dam (Wang et al., 2011a). These measures have caused the Loess Plateau to experience a substantial change in land use, vegetation cover, soil properties, and catchment hydrology (Chen et al., 2007; Wang et al., 2011b; Wei et al., 2014). Consequently, the hydrological and carbon biogeochemical processes, which operate and interact with each other, were dramatically altered (Liang et al., 2015a; Liang et al., 2015b). These changes in hydrology and soil carbon cycle induced by land use and vegetation change may particularly important in the dynamics of DOC concentration and flux in an ecological restored catchment. Moreover, the majority of annual rainfall is concentrated between July and September in LPR. Less information is available on DOC export driven by rainfall event, which DOC flux is an important component in overall carbon balance for ecological restored

catchment.

Therefore, the primary goal of this study is to investigate how variations of DOC concentration and flux response to a sequence of rainfall events from a restored catchment during concentrated rainfall season in the LPR. Specifically, the two objectives of this study were (1) to examine the dynamic changes in DOC concentration and flux and assess the difference in DOC export driven by various rainfall events, and (2) evaluate how rainfall, runoff, and antecedent factors affect DOC export from a catchment. To do so, we used high-frequency method to capture the temporal changes in DOC export and hydrological process driven by rainfall event within an ecological restored watershed in LPR. These results will provide evidence of DOC export response to rainfall events, especially driven by extreme events, which may be important for evaluating carbon balance and modeling DOC export through runoff at ecological restored catchment in LPR.

**Comment 4.** The three objectives of this study were not well described in the introduction section.

**Response:** Thanks for your suggestions. The objectives have been reorganized.

**Line 73-76:** Specifically, the two objectives of this study were (1) to examine the dynamic changes in DOC concentration and flux and assess the difference in DOC export driven by various rainfall events, and (2) evaluate how rainfall, runoff, and antecedent factors affect DOC export from a catchment. To do so, we used high-frequency method to capture the temporal changes in DOC export and hydrological process driven by rainfall event within an ecological restored watershed in LPR.

**Comment 5.** The result section is too long, making it difficult to read.

Response: Thanks for your suggestions. The result section has been reorganized in Line 152-222 as following:

[revised manuscript text omitted]

**Comment 6.** The authors need to redo the tables and figures. I do not understand Figure 2a. It is also difficult to understand Fig. 8. The authors also need to explain the abbreviations for R1, R2 in table 2 so the readers need not to find them in the text.

Response: Thanks for your suggestions. We redo the Table 1, 2 and Figure 2 a as following:

**Table 1**

| Date | Ra(mm) | Flow rate (L s⁻¹) | $C_f$ (mg L⁻¹) | *Flux* (kg km⁻²) | Date | Ra(mm) | Flow rate (L s⁻¹) | $C_f$ (mg L⁻¹) | *Flux* (kg km⁻²) |
|---|---|---|---|---|---|---|---|---|---|
| 1-Jun. | 1.0 | 0.56 | 10.87 | 0.52 | 1-Aug. | 0.8 | 0.54 | 5.14 | 0.24 |
| 2-Jun. | 7.0 | 0.59 | 9.97 | 0.51 | 2-Aug. | 4.2 | 0.63 | 9.72 | 0.53 |
| 3-Jun. | 3.0 | 0.53 | 10.53 | 0.48 | 6-Aug. | 0.8 | 0.47 | 7.95 | 0.32 |
| 5-Jun. | 3.2 | 0.53 | 11.59 | 0.53 | 12-Aug. | 0.8 | 0.40 | 5.30 | 0.18 |
| 7-Jun. | 13.8 | 0.65 | 12.96 | 0.73 | 13-Aug. | 1.2 | 0.35 | 5.93 | 0.18 |
| Jun. | 82.9 | 0.35 | 11.52 | 102.39 | 16-Aug. | 18.8 | 0.96 | 6.46 | 0.54 |
| 11-Jul. | 24.6 | 0.44 | 11.92 | 0.45 | 17-Aug. | 0.6 | 0.55 | 9.69 | 0.46 |
| 13-Jul. | 19.8 | 1.28 | 11.84 | 1.31 | 18-Aug. | 1.2 | 0.57 | 7.44 | 0.37 |
| 14-Jul. | 11.0 | 0.46 | 13.00 | 0.52 | Aug. | 53.8 | 0.53 | 6.81 | 94.73 |
| 18-Jul. | 62.6 | 1.46 | 11.64 | 1.47 | 9-Sept. | 6.8 | 0.44 | 13.14 | 0.50 |
| 19-Jul. | 29.2 | 4.05 | 8.12 | 2.84 | 10-Sept. | 21.8 | 1.24 | 7.21 | 0.77 |
| 31-Jul. | 2.2 | 0.54 | 6.70 | 0.31 | 17-Sept. | 6.2 | 0.48 | 9.10 | 0.38 |
| Jul. | 184.2 | 0.41 | 8.95 | 96.57 | Sept. | 51.2 | 0.57 | 7.49 | 110.17 |

**Table 2**

| | *Flux* | Q | Ra | R1 | R7 | R14 | REI | $T_{air}$-7 | $T_{air}$-14 | SMC-7 | SMC-14 |
|---|---|---|---|---|---|---|---|---|---|---|---|
| $C_f$ | 0.30 | -0.01 | 0.30 | -0.01 | 0.23 | -0.05 | -0.32* | -0.25 | -0.24 | -0.44** | -0.65** |
| *Flux* | | 0.94** | 0.69** | 0.76** | 0.57** | 0.29 | -0.14 | -0.07 | -0.04 | 0.06 | -0.24 |
| Q | | | 0.60** | 0.85** | 0.53** | 0.33* | -0.07 | -0.02 | 0.01 | 0.19 | -0.03 |
| Ra | | | | 0.38* | 0.39* | 0.14 | -0.06 | 0.02 | 0.07 | -0.05 | -0.30 |
| R1 | | | | | 0.58** | 0.42** | -0.27 | 0.11 | 0.10 | 0.12 | -0.01 |
| R7 | | | | | | 0.69** | -0.28 | 0.24 | 0.23 | 0.40** | 0.02 |
| R14 | | | | | | | -0.20 | 0.19 | 0.13 | 0.56** | .420** |
| REI | | | | | | | | -0.02 | 0.03 | 0.26 | 0.25 |
| $T_{air}$-7 | | | | | | | | | 0.96** | 0.09 | 0.20 |
| $T_{air}$-14 | | | | | | | | | | 0.09 | 0.17 |
| SMC-7 | | | | | | | | | | | 0.79** |

Note: ** ($P<0.01$), * ($P<0.05$).

$C_f$: Flow-weighted mean concentration driven by an event, Flux: Event-driven DOC quantity,

Q: Total discharge volume, Ra: Total rainfall amount,

R1: Total rainfall amount in the 1 day before the current rainfall event,

R7: Total rainfall amount in the 7 days before the current rainfall event,

R14: Total rainfall amount in the 14 days before the current rainfall event,

SMC-7 and SMC-14: Soil moisture content in the 7 and 14 days before the current rainfall event,

$T_{air}$-7 and $T_{air}$-14: Mean air temperature in the 7and 14 days before the current rainfall event,

REI: Interval days between the current and last rainfall event.

**Figure 2**

[Figure]

**Comment 6.** The authors should clarify the rainfall amount and rainfall intensity, which is important to class the rainfall events.

**Response:** Thanks for your suggestions. Indeed, the rainfall events were grouped by rainfall amount and the Figure 3 has been changed. According to Yang's results, the threshold of rainfall amount mean rainwater can effectively recharge the soil water in LPR, which may affect soil moisture content. This is why we selected this classification. Thus, we choose the parameter of rainfall amount to analyze in this manuscript.

**Line 156-158:** All the rainfall events in between June to September were grouped into four grades: <5 mm (Light rainfall), 5-10 mm (Moderate rainfall), 10-20 mm (Heavy rainfall), and >20 mm (Violent rainfall) according to rainfall amount classification (Yang et al., 2018).

**Figure 3:**

[Figure]

**Comment 7.** it is unclear about the time interval between these sampling times. If they want to conduct such analysis, they should check the original data to ensure the normal distribution.

**Response:** Thanks for your suggestions. details about sampling times showed in method part in Line 102-104. The regression was removed due to lack of data normal distribution and Figure 4 also has been changed accordingly.

**Line 102-104:** High-frequency monitoring was carried out in a rainfall event based hydrological process, thus the ISCO was set to acquire samples every 10 min from the first 12 runoff samples and another 12 were sampled every 30 min.

**Figure 4**

[Figure]

**Comment 8.** Conclusions. The findings of this study indicate that DOC concentrations were highly variable, particularly during low runoff discharge periods, granted, this belongs to the conclusion. The authors should think hard about the findings of this study and show that these findings are valuable.

**Response:** Thanks for your suggestions. The conclusion part has been rewritten and the details show in Line 327-342 of this manuscript.

**Line 316-331:** The DOC concentration and flux for individual rainfall events from a semi-arid catchment of the LPR was initially monitored during the concentrated rainfall season. DOC concentration showed a weak correlation with discharge, except in higher runoff discharge induced by extreme rainfall events. The findings of this study indicate that DOC concentrations were highly variable, particularly during low runoff discharge periods. Hysteresis analysis showed that the relationship between DOC concentration and runoff discharge for a rainfall event is nonlinear and varied with conditions in rainfall amount, discharge process. DOC flux increased with runoff discharge and showed a positive linear correlation with runoff discharge. These results showed that higher DOC flux with low DOC concentration related to higher discharge and its dilution effects in a hydrological process driven by larger rainfall amount. The diluted DOC concentration induced by increased discharges contributed slightly to difference in DOC flux, due to total runoff discharge is a major variable for flux. These results showed that the temporal variation magnitude of DOC is related to hydrological condition (Q and Ra) and antecedent condition (R1, R7 and SMC), and suggested that the event-driven DOC export is largely influenced by rainfall through direct effects on catchment hydrology and indirect effects on soil carbon cycles. Changes in catchment hydrology and soil carbon processes responded to climate change may play an important role in terrestrial carbon export, in particular for a restored catchment. Thus, further work should focus on carbon export response to various rainfall events at a larger spatiotemporal scale for better estimating future terrestrial carbon flux to aquatic ecosystem and evaluating carbon balance in ecological restored catchment in LPR. In addition, engineers and scientists can take advantage of the derived results to better develop advanced field monitoring work.

**Other Changes:**

We have revised the abstract part in Line 11-26. We also added discussion information in Line 261-271 and Line 304-314. The details show in the following part:

In Abstract **Line 11-26:** Dissolved organic carbon (DOC) transported by runoff has been identified as an important role of the global carbon cycle. Despite there being many studies on DOC concentration and flux, but little information is available in semi-arid catchments of the Loess Plateau Region (LPR). The primary goal of this study was to quantify DOC exported driven by a sequence of rainfall events during the concentrated rainfall season. In addition, factors that affect DOC export from a small headwater catchment will be investigated accordingly. Runoff discharge and DOC concentration were monitored at the outlet of the Yangjuangou catchment in Yanan, Shaanxi Province, China. The results showed that DOC concentration was highly variable, with event-based DOC concentrations ranging from 4.08 to 15.66 mg L$^{-1}$. Hysteresis analysis showed a nonlinear relationship between DOC concentration and flow rate in the hydrological process. The monthly DOC flux loading from the catchment was 94.73-110.17 kg km$^{-2}$ from June to September, while the event-based DOC flux ranged from 0.18 to 2.84 kg km$^{-2}$. Variations of event-driven DOC concentration contributed slightly to a difference in DOC flux, whereas intra-events of rainfall amount and runoff discharge led to evident difference in DOC export. In conclusion, our case results highlighted the advantages of high-frequency monitoring for DOC export and indicated that event-driven DOC export is largely influenced by the interaction of catchment hydrology and antecedent condition within a catchment. Engineering and scientists can take advantage of the derived results to better develop advanced field monitoring work. In addition, more studies are needed to investigate the magnitude of terrestrial DOC export in response to projected climate change at larger spatiotemporal scale, which may have implication for the carbon balance and carbon cycle model from an ecological restored catchment in LPR.

**Line 261 -271:** For event-driven flux, the DOC flux is a function of total runoff discharge and DOC concentration ($C_f$). DOC flux showed a positive linear relationship with runoff discharges, which is not surprising and parallel with studies reported by Clark et al. (2007) and Ma et al. (2018). In addition, it should be noted that the DOC flux induced by larger rainfall amount was higher than flux driven by light rainfall, whereas the $C_f$ showed no evident difference for the selected rainfall events. Thus, the greater DOC flux clearly showed that the DOC export was close linked to hydrologic process induced by various amount of rainfall event in LPR. For an ecological restored catchment in LPR, the soil carbon driven by increased vegetation was significantly increased and acted as a positive pathway to sequestration soil carbon on terrestrial ecosystem (Wang et al., 2011b). Meanwhile, the reduced hydrology responded to an increased vegetation may diminish soil carbon transported by hydrological process in a catchment. The event-driven DOC transport is an important component for evaluating carbon balance of the ecological restored catchment in LPR. Hence, further study should be long-term undertaking to investigate the hydrological response and its impact on terrestrial carbon loss from a catchment in LPR.

**Line 304-314:** DOC flux was significantly and positively correlated with Q, Ra, R1 and R7. The Q

and Ra reflect the direct effect of current rainfall and hydrological processes during a rainfall event, while R1 and R7 refer to the antecedent rainfall conditions and reflect indirect effects on DOC export. These results agreed with previous studies demonstrated by Blaen et al. (2017), who noted that antecedent conditions and rainfall were key drivers of DOC export during a rainfall event. Cooper et al. (2007) also concluded that DOC export is largely governed by interactions between hydrological and meteorological factors and carbon biogeochemical process. Overall, these results suggested that rainfall is a key factor influencing hydrological process, and thus DOC export from an ecological restored catchment in LPR. Apart from the increased soil carbon driven by increased vegetation (Wang et al., 2011b), the weaken hydrological process induced by increased vegetation may also cause a less terrestrial carbon export from a catchment. Therefore, our results highlight the need for research not only into the hydrological process and soil carbon cycle, but the integration of carbon export driven by a sequence of rainfall events across spatiotemporal scales to understand the carbon balance in a restored catchment in LPR.

---

## Author Comment (AC4) · 11 Apr 2019

[revised manuscript text omitted]

---

## Author Comment (AC5) · 11 Apr 2019

The comment was uploaded in the form of a supplement:
https://www.hydrol-earth-syst-sci-discuss.net/hess-2019-8/hess-2019-8-AC5-supplement.pdf

---

## Author Comment (AC6) · 11 Apr 2019

The comment was uploaded in the form of a supplement:
https://www.hydrol-earth-syst-sci-discuss.net/hess-2019-8/hess-2019-8-AC6-supplement.pdf

---

## Author Response (AR2)

**Dear Editor and Referees:**

Thank you for your letter and for the referees' comments concerning our manuscript. We have carefully revised our manuscript after reading the comments. Revised portion are marked in red in this response letter. The point-by-point corrections in the manuscript according to the referees' comments are as following:

**Response to Referee #1**

**Comment 1.** There are some repetitions of phrases in some sentences, for example, page 33 line 880-881 and page 34 line 931-932.

**Response:** Thank you for your comments. The Green Line in page 34 (line 931-932) means the sentences were
10     deleted. Sorry for this error and we've checked again in this updated manuscript.

**Comment 2.** Page 34 line 910-912. Did the studies of Blaen et al. (2017) and Clark et al. (2007) show the same DOC concentration range? Please check if the referred data is accurate.

**Response:** Thank you for your comment. We've checked the data again in this updated manuscript.

15     **Line 47-49:** For instance, Clark et al. (2007) found that DOC concentration varied from 5 to 35 mg L$^{-1}$ with a highly variable in rainfall events from a peatland catchment, and a study by Blaen et al. (2017) showed that the DOC concentration ranged from 5.4 to 18.9 mg L$^{-1}$.

**Comment 3.** Page 34 line 933, page 35 line 951, page 42 line 1222, page 43 line 1247 and page 44 line 1277 and
20     1280. Please check and correct the grammar or spelling of those sentences.

**Response:** Thank you for your comments. We've revised the grammar and spelling in this updated manuscript.

**Line 260-261:** A figure-of-eight hysteresis was observed in 2-August due to the DOC concentration keep pace with flow rate during the rising and falling limb.

**Line 268-270:** Thus, the greater DOC flux clearly showed that the DOC export was closely linked to hydrologic
25     process induced by various amount of rainfall events in LPR.

**Line 289-293:** Positive correlation between Ra, R7 and $C_f$ suggested that the combination of the current rainfall amount and the accumulated rainfall before a current rainfall event played important roles in DOC concentration for a rainfall event. R7 may reflect the antecedent hydrological condition and Ra represent the current rainfall input into the catchment. Higher Ra and R7 may lead to a well hydrological connectivity, and thus more DOC source may
30     contribute to runoff.

**Comment 4**. Page 41 line 1166. I think it should be Figure 6-c instead of Figure 6-2.

**Response:** Sorry for the error. We've revised the figure number in this updated manuscript.

**Line 209-210:** A figure-of-eight pattern and indicated that $C_i$ generally varied in pace with runoff discharge on
35     2-August, 2016 (Figure 6-c).

**Comment 5**. Page 49 line 1469 and line 1471. The date in the figure title does not match that in text and figures. Do the 6-June (page 42 line 1213 and 1215) and 7-June (page line 1142) rainfall events refer to the same rainfall event? Please check and unify the date of rainfall event in June.

40     **Response:** Sorry for the mistakes. We've corrected the date in this updated manuscript.

**Line 252-254:** In our result, the anticlockwise hysteresis between DOC concentration and flow rate was observed at 7-June. The peak DOC concentration was delayed compare to peak flow rate. These results may be attributed to a 5.2 mm rainfall was happen earlier than the maximum rainfall at 7-June (Figure 5-a).

**Line 458-461:** Figure 5 Dynamic changes of DOC concentration ($C_i$) in an individual runoff event: (a)7-June, (b)13-July, (c)2-August, (d)10-September.

Figure 6 Hysteresis loops for four selected runoff events from June to September: (a)7-June, (b)13-July, (c)2-August, (d)10-September.

**Comment 6**. Page 51. The column names in Table 2 need to be modified to be consistent with the text and table notes.

**Response:** Thank you for your comment. We've revised Table 2 accordingly.

**Table 2**

|  | *Flux* | Q | Ra | R1 | R7 | R14 | REI | $T_{air}$-7 | $T_{air}$-14 | SMC-7 | SMC-14 |
|---|---|---|---|---|---|---|---|---|---|---|---|
| $C_f$ | 0.30 | -0.01 | 0.30 | -0.01 | 0.23 | -0.05 | -0.32* | -0.25 | -0.24 | -0.44** | -0.65** |
| *Flux* |  | 0.94** | 0.69** | 0.76** | 0.57** | 0.29 | -0.14 | -0.07 | -0.04 | 0.06 | -0.24 |
| Q |  |  | 0.60** | 0.85** | 0.53** | 0.33* | -0.07 | -0.02 | 0.01 | 0.19 | -0.03 |
| Ra |  |  |  | 0.38* | 0.39* | 0.14 | -0.06 | 0.02 | 0.07 | -0.05 | -0.30 |
| R1 |  |  |  |  | 0.58** | 0.42** | -0.27 | 0.11 | 0.10 | 0.12 | -0.01 |
| R7 |  |  |  |  |  | 0.69** | -0.28 | 0.24 | 0.23 | 0.40** | 0.02 |
| R14 |  |  |  |  |  |  | -0.20 | 0.19 | 0.13 | 0.56** | .420** |
| REI |  |  |  |  |  |  |  | -0.02 | 0.03 | 0.26 | 0.25 |
| $T_{air}$-7 |  |  |  |  |  |  |  |  | 0.96** | 0.09 | 0.20 |
| $T_{air}$-14 |  |  |  |  |  |  |  |  |  | 0.09 | 0.17 |
| SMC-7 |  |  |  |  |  |  |  |  |  |  | 0.79** |

Note: ** (P<0.01), * (P<0.05).

$C_f$: Flow-weighted mean concentration driven by an event, *Flux*: Event-driven DOC quantity,

Q: Total discharge volume, Ra: Total rainfall amount,

R1: Total rainfall amount in the 1 day before the current rainfall event,

R7: Total rainfall amount in the7days before the current rainfall event,

R14: Total rainfall amount in the14 days before the current rainfall event,

SMC-7 and SMC-14: Soil moisture content in the 7 and 14 days before the current rainfall event,

$T_{air}$-7 and $T_{air}$-14: Mean air temperature in the 7and 14 days before the current rainfall event,

REI: Interval days between the current and last rainfall event.

**Response to Referee #2**

65 **Comment 1:** Is there absolutely no data on DOC in this region? If DOC has been reported in the region, what kind of information is available? What did those tell us?

**Response:** Thank you for your comments. Our work is a good study that point out the need of more scientifically credible data here in the future. There is less data on DOC in this region and line 49-50 described a study in this region.

70 **Line 49-50:** Similar results were reported by Ran et al. (2018), who found that DOC concentration ranged from 1.4 to 9.5 mg L$^{-1}$ in the Wuding River in the LPR.

**Comment 2:** why people don't care about DOC in the study area before? What didn't previous studies show?

**Response:** Thank you for your comments. It is our understanding that there are also some possible practical reasons
75 why people didn't care about DOC in hydrological process in the study area. We've added some more information as following:

**Line 71-72:** Moreover, DOC transport from a catchment is sparsely measured due to the DOC concentration in hydrological process is not treated as a general parameter in monitoring networks of LPR.

**Line 104-109:** Unlike the common sampling frequency is monthly or weekly at field observatory station, the
80 high-frequency monitoring was carried out in a hydrological process driven by a rainfall event in this study. Researchers resided in the field observatory station and treated the samples immediately after a rainfall event to ensure that the DOC in the sampled water did not microbially degrade (Kieber et al., 2002; Willey et al., 2000). The time-consuming and laborious field work is also one of the reasons for the measurement scarcity of DOC export in the existing ecosystem monitoring networks.

85

**Comment 3:** And what changed make you start to care?

**Response:** Thank you for your comment. The Line 68-73 described why we start to focus on DOC transport in this study region.

[revised manuscript text omitted]